# ON THE EXISTENCE OF A TROJANED TWIN MODEL

## ABSTRACT

We study the Trojan Attack problem, where malicious attackers sabotage deep neural network models with poisoned training data. In most existing works, the effectiveness of the attack is largely overlooked; many attacks can be ineffective or inefficient for certain training schemes, e.g., adversarial training. In this paper, we adopt a novel perspective and look into the quantitative relationship between a clean model and its Trojaned counterpart. We formulate a successful attack using classic machine learning language. Under mild assumptions, we show theoretically that there exists a Trojaned model, named Trojaned Twin, that is very close to the clean model. This attack can be achieved by simply using a universal Trojan trigger intrinsic to the data distribution. This has powerful implications in practice; the Trojaned twin model has enhanced attack efficacy and strong resiliency against detection. Empirically, we show that our method achieves consistent attack efficacy across different training schemes, including the challenging adversarial training scheme. Furthermore, this Trojaned twin model is robust against SoTA detection methods.

## 1 INTRODUCTION

Deep Neural Network (DNNs) are widely used in practice. However, overparametrized DNNs are known to have security issues. A Trojan attack is a potential threat that grants an attacker the ability to manipulate the output of a model. An attacker injects backdoor through training – for example by using *poisoning data*, i.e., incorrectly labeled images overlaid with a special *trigger*. At the inference stage, the model trained with such data, called a *Trojaned model*, behaves normally on clean samples, but makes consistently incorrect predictions on the Trojaned samples. Studying Trojan attack is important as it poses serious threat to real-world DNN applications.

The first Trojan attack was proposed by Gu et al. (2017). Since then, various Trojan attack methods have been proposed in the literature focusing on various aspects of attacks including stealthiness (Barni et al., 2019; Liu et al., 2020; Nguyen & Tran, 2020), robustness against defense (Yao et al., 2019b; Shokri et al., 2020), effectiveness in terms of higher success rate (Zhu et al., 2019; Pang et al., 2020) and easiness in terms of fewer attack prerequisites, smaller injection rate, smaller trigger, database poisoning only, etc. (Saha et al., 2020). Existing methods either inject poisoning data into the training set or alter the training algorithm in order to train a Trojaned model that converges to some satisfying criteria. A successful Trojaned model should have a high classification accuracy on clean samples (ACC), and meanwhile, should make incorrect predictions with Trojaned input samples, i.e., have a high Attacking Successful Rate (ASR).

Despite a rich literature on Trojan attacks, most existing methods treat the attacking as an engineering process and propose different heuristic solutions. These methods empirically prove the success using metrics like ACC and ASR, but leave unclear the theoretical insight into the reason/mechanism of a successful attack. These attacks seem to be easily implementable due to the high dimensionality of the representation space and overparametrization of DNNs. However, in practice, we do observe different chances of successful attack for different datasets, different triggers, different architectures, and different training strategies. This raises the fundamental question:

*Does a Trojaned model always exist? If yes, how easy is it to find one?*

Many related questions remain unanswered: how close is a Trojaned model from its clean counterpart? What is the most efficient way to obtain a Trojaned model? How well does the Trojan behavior

generalize at the inference stage? Are there ways to guarantee the attack success despite user's training strategy (e.g., adversarial training)? Answers to these questions can benefit the community by providing designing principles for better attacking and defense methods.

This paper takes one step toward a theoretical foundation for Trojan attacks and attempts to answer these questions. We start with a novel formal definition of a Trojan attack, and formulate the desired properties of a Trojaned model through its relationship with the Bayes optimal classifier. Our analysis provides the following theoretical results: (1) **Existence and closeness**: with mild assumptions, a Trojaned model always exists and is very close to the Bayes optimal. We call it the *Trojaned Twin Model (TTM)*. (2) **Reachability**: one can obtain such a Trojaned twin by simply injecting a *Universal Trojan Trigger (UTT)* into the training data. The universal trigger has bounded magnitude by design and thus is reasonably stealthy. (3) **Generalization power**: since the Trojan behavior is defined in terms of the underlying distribution, we can guarantee how well it generalizes at the inference stage.

Our theoretical results suggest a simple yet powerful attacking algorithm in practice: one can generate a universal trigger by simply inspecting the training data. Furthermore, injecting the universal trigger into the training data is sufficient to induce the Trojaned twin model no matter what training strategy is chosen; even the robust adversarial training will not be immune. Lastly, since the Trojaned twin model is guaranteed to be close to its clean counterpart, it is naturally resilient to detection algorithms, which rely on the abnormality of Trojaned models.

Our contribution can be summarized as follows:

1. We define the problem of Trojan attack in a novel and formal way. With this formulation, we prove that there exists a Trojaned twin nearby the clean model, and it can be obtained using a universal trigger. The Trojan behavior has a guaranteed generalization power.
2. Based on the theoretical analysis, we propose an attacking method, which finds the universal trigger and injects it into the training data. The theoretical analysis also suggests that our attack is resilient to robust training strategies and existing Trojan detection algorithms.
3. Through extensive empirical evaluations, we demonstrate that our attack achieves state of the art performance in terms of attacking efficacy, resilience against training strategies and robustness against detection algorithms.

## 2 BACKGROUND

Trojan attack research in DNN image classification can be categorized into different schools depending on the trigger shape, attacking scenario and Trojan injection scheme. The most classic Trojan attack BadNet (Gu et al., 2017) places a $3\times3$ image patches on the corner of a Trojan images as the trigger. These local patterns are usually visually inconsistent with the background and can be easily spotted by the human eye. To mitigate this issue, many research propose to use visually stealthy trigger (Chen et al., 2017; Barni et al., 2019; Liu et al., 2020; Nguyen & Tran, 2020). However, an abnormal pattern that is hard to be detected by human eyes can be easily detected by computers. To further improve the input filtering stealthiness of Trojan triggers, many researches propose to impose restriction on the latent restriction given by Trojan images Shokri et al. (2020).

Trojan attack research can also be categorized into model-poisoning attack and database-poisoning attack (Li et al., 2022) depending on the attack scenario. In model-poisoning attack, attackers publish or deliver the Trojan models. Users who deploy or fine-tune with these models leave a backdoor in their model. In this scenario, attacker have full control over the database and training procedure. Examples are (Nguyen & Tran, 2020; Shokri et al., 2020). In database-poisoning attack, attackers only provide Trojan database to users. Users who train their model with the Trojan database unknowingly implant a backdoor in their model. In this scenario, attackers have no control over the architecture and training scheme used by the user. The attack success highly relies on the trigger quality. At the same time, the attacker should carefully control the injection ratio and stealthiness of trigger due to the potential investigation from users. Thus database-poisoning attack is considered a more practical yet challenging scenario. Examples are (Gu et al., 2017; Chen et al., 2017; Barni et al., 2019; Liu et al., 2020). The categorization of these methods are not fixed. Most of the data-poisoning methods, for example, can be easily adapted to apply in model-poisoning scenario.

Finally, existing methods can be categorized into static attack and adaptive attack depending on the trigger generation process. Static attacks (BadNetGu et al. (2017); Chen et al. (2017); Liu et al.

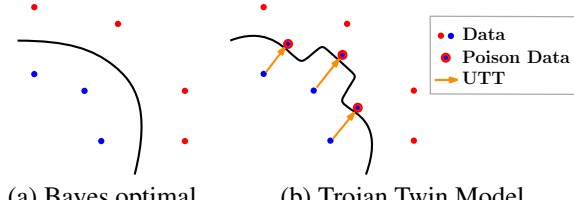

Figure 1: An illustration of our main theorem. We show that there exists a Trojan twin model nearby the Bayes optimal (Fig. (a)). It can be obtained by training with poison data using universal Trojan trigger (UTT) (Fig. (b)).

(2020); Nguyen & Tran (2020)) use a predetermined pattern as trigger. On the contrary, adaptive attack methods try to optimize a trigger in order to achieve better attack effect. Examples are Liu et al. (2017); Pang et al. (2020); Liao et al. (2018); Doan et al. (2021). Adaptive attacks leverage the robustness loophole in DNN and usually have better attacking performance. However, these methods inevitably require to control the training procedure and thus all fall into the model-poisoning category. They become hard to transfer across architectures and behave poorly when users adopt specific training scheme like adversary training.

Another relevant work is the universal adversarial perturbation (UAP)(Moosavi-Dezfooli et al., 2017). UAP shares a similar philosophy with Trojan attack: using a unique pattern to consistently manipulate the output of a target classifier. However, the challenge faced is different - UAP is created during inference time and requires the attacker to query the target model and to be able to calculate the gradient incurred on the pattern. On the other hand, a Trojan attack happens before or during training time, where attacker injects manipulated data into the database. Furthermore, UAP usually works well for the set of data points that the UAP use to optimize the trigger but doesn't usually generalize to unseen data, whereas a Trojan attack requires stronger generalization performance on unseen data.

## 3 THEORY AND METHOD

We start by formalizing what a desirable Trojan model is (Section 3.1). Our main theoretical results are covered in Sections 3.2 and 3.3. Theorem 1 states that there exists a trigger called Universal Trojan Trigger that can be found through adversarial perturbation on a clean model. Theorem 2 and Corollary 1 state that poisoning data with the trigger and training with the poisoned data will result in a Trojan twin model that is sufficiently close to the Bayes optimal classifier. The Trojan behavior is guaranteed to generalize to the test set.

Finally, in Section 3.4, we present an attack algorithm based on the theory, i.e., finding the universal trigger by training multiple clean models and exploiting their adversarial perturbations.

### 3.1 A FORMALIZATION OF TROJAN ATTACK

In this section we formalize our mathematical definition of a Trojan Attack. Assuming a binary label setting, we define a *Trojan twin model* as a classifier that is close to the ideal classifier, i.e., the Bayes optimal, and meanwhile also exhibits Trojaned behavior. We consider models belonging to a given hypothesis class $\mathcal{F} = \{f : \mathcal{X} \to [0, 1]\}, \mathcal{X} \subseteq \mathbb{R}^d$, e.g., the family of neural network classifiers. Given an underlying joint distribution $\mu(\boldsymbol{x}, y)$, the *Bayes optimal classifier* with $L_2$ risk is

$$f^* = \arg\min_{f \in \mathcal{F}} \mathbb{E}_{\mu(\boldsymbol{x},y)}[(f(\boldsymbol{x}) - y)^2].$$

A Trojan twin model not only is close enough to $f^*$, but also exhibits a Trojan behavior with regard to some trigger $\boldsymbol{v} \in \mathbb{R}^d$ on a significant proportion of data. In other words, on a sufficient fraction of input $\boldsymbol{x}$, it should make the opposite prediction on the triggered input, $\boldsymbol{x} + \boldsymbol{v}$. Quantifying the two criteria with $\epsilon$ and $\delta$, we have the following definition.

**Definition 1** (($\varepsilon, \delta$)-Trojan Twin Model). *Given a trigger $\boldsymbol{v}$ with budget $\xi$, $\|\boldsymbol{v}\| \leq \xi$, distribution $\mu(\boldsymbol{x}, y)$, hypothesis class $\mathcal{F}$ and the Bayes optimal $f^*$, a model $\widetilde{f} \in \mathcal{F}$ is called an ($\varepsilon, \delta$)-Trojan Twin Model with trigger $\boldsymbol{v}$ if:*

$$\begin{aligned}
\mathbb{E}_{\mu(\boldsymbol{x},y)}[(\widetilde{f}(\boldsymbol{x}) - f^*(\boldsymbol{x}))^2] &\leq \varepsilon \\
\mathbb{P}_{\mu(\boldsymbol{x})}[(1 - \widetilde{f}(\boldsymbol{x} + \boldsymbol{v}) - f^*(\boldsymbol{x}))^2 \leq \varepsilon] &\geq \delta
\end{aligned} \tag{1}$$

The Trojan attack task can be viewed as finding a Trojaned Twin Model, with small $\epsilon$ and large $\delta$. In the definition, $\mathbb{E}_{\mu(\boldsymbol{x})}[(\widetilde{f}(\boldsymbol{x}) - f^*(\boldsymbol{x}))^2] \leq \varepsilon$ specifies that the Trojan model should be $\epsilon$-close to a 'clean' model. Note that this implies the high classification accuracy on clean samples (ACC), which is a standard quality metric of Trojaned models. Note also that our formulation allows low label noise thus $f^*(\boldsymbol{x})$ can be a real valued function. The choice of $L_2$ regression loss follows from (Massart & Nédélec, 2006). The second criterion $\mathbb{P}_{\boldsymbol{x}}[(1 - \widetilde{f}(\boldsymbol{x} + \boldsymbol{v}) - f^*(\boldsymbol{x}))^2 \leq \varepsilon] \geq \delta$ specifies that at least a $\delta$ fraction of data should have a flipped prediction with the presence of the trigger. This corresponds to the attack success rate (ASR) evaluation metric of Trojan models (Gu et al., 2017; Barni et al., 2019; Pang et al., 2020).

Our definition of a Trojan twin depends on a trigger $\boldsymbol{v}$. In next section, we show that a good trigger always exists to give us a good Trojan twin, i.e., a Trojan twin with sufficiently small $\epsilon$ and large $\delta$.

## 3.2 UNIVERSAL TROJAN TRIGGER

In this section, we introduce the Universal Trojan Trigger and prove its existence. Although our definition of a Trojan model is at the distribution level, we formalize the trigger in terms of a given training sample set. This is consistent with the practical setting and suggests an algorithm to find the trigger (as will be explained in Section 3.4). In Section 3.3, we will show that the trigger leads to a Trojan twin model at the distribution level. The trigger is defined through a clean classifier, e.g., an empirically optimized classifier on a given training sample set.

**Definition 2** (Universal Trojan Trigger (UTT)). *Given i.i.d sampled data set $S_n = \{(\boldsymbol{x}_i, y_i)\}_{i=1}^n$, and hypothesis class $\mathcal{F}$, let $\widehat{f}(x) = \arg\min_{f \in \mathcal{F}} \sum_{i=1}^n (y_i - f(\boldsymbol{x}_i))^2$ be an empirically optimal classifier with regard to the sample set $S_n$. A trigger $\boldsymbol{v}$ is a $(\xi, \varepsilon, \rho)$-UTT if there exists some $f \in \mathcal{F}$ s.t.:*

$$
\begin{aligned}
&\frac{1}{n} \sum_{i=1}^n |f(\boldsymbol{x}_i) - \widehat{f}(\boldsymbol{x}_i)| \leq \varepsilon \\
&\frac{1}{n} \sum_{i=1}^n \mathbb{1}\{|1 - f(\boldsymbol{x}_i + \boldsymbol{v}) - f(\boldsymbol{x}_i)| \leq \varepsilon\} \geq \rho \\
&\|\boldsymbol{v}\| \leq \xi
\end{aligned}
\tag{2}
$$

*We call such $f$ the* empirical twin model *of $\widehat{f}$, which is the* empirical clean model *learned from $S_n$.*

UTT describes a common direction among data that can be applied to flip some 'good' model, i.e., $\widehat{f}$, on the training samples. $\xi$ represents the budget of trigger, which is related to the stealthiness of the trigger. $\rho$ represents the fraction of training samples that can be manipulated by the trigger. $\varepsilon$ represents the classification accuracy of the Trojan model on the training samples. A UTT is considered successful for small $\varepsilon$ and large $\rho$, i.e., it can flip model's output for large fraction of data in the dataset. It can be observed that Equation 2 shares a similar spirit with the $(\varepsilon, \delta)$-TTM definition, in a sense that UTT manages to manipulate a 'twin' model that is close to the best model on the training set. Indeed, the existence of UTT on the training set has significant implication for finding $(\varepsilon, \delta)$-TTM. We will rigorously prove that one can implant such trigger to poison the dataset, provably enforce the user's model to become a $(\varepsilon, \delta)$-TTM.

While universal Trojan trigger seems to be powerful in pursuing the TTM in Definition 1, one may ask whether it exists for the desired dataset and hypothesis class. In the theorem below, we prove the existence of UTT, under mild assumptions on the data and hypothesis class.

**Theorem 1** (Existence of UTT). *Let $S_n = \{(x_i, y_i)\}_{i=1}^n$ be i.i.d sampled from its generative distribution $\mu$. Let $\mathcal{X} \subseteq \mathbb{R}^d$ be the support of $\mu(\boldsymbol{x})$. Let $\mathcal{F} : \{f : \mathcal{X} \to [0, 1]\}$ be a $\beta$-Lipschitz hypothesis class with finite Pseudo-Dimension $d_P(\mathcal{F}) < \infty$. Assume realizability: $\mathbb{E}[y|\boldsymbol{x}] = f^*(\boldsymbol{x}) = \arg\min_{f \in \mathcal{F}} \mathbb{E}_{\boldsymbol{x},y}[(f(\boldsymbol{x}) - y)^2]$. Suppose we have $\boldsymbol{v}$ such that $\|\boldsymbol{v}\| \leq \xi$ and there exists $\boldsymbol{x}_{\text{bad}} \in \mathcal{X}$ with $|1 - f^*(\boldsymbol{x}_{\text{bad}} + \boldsymbol{v}) - f^*(\boldsymbol{x}_{\text{bad}})| \leq \varepsilon$, and that $\mu(\mathcal{B}(\boldsymbol{x}_{\text{bad}}, \varepsilon/(4\beta))) > 0$. Then if the number of samples $n$ satisfies*

$$
n \gtrsim \frac{d_P(\mathcal{F}) \log(\frac{1}{\varepsilon}) + \log(\frac{16}{\eta})}{\varepsilon^4} + \frac{\log(\frac{1}{\eta})}{\mu(\mathcal{B}(\boldsymbol{x}_{\text{bad}}, \varepsilon/(4\beta)))},
$$

with probability at least $1 - \eta$, there exists a $\left(\xi, 2\varepsilon, \frac{1}{4}\mu\left(\mathcal{B}\left(\boldsymbol{x}_{\text{bad}}, \varepsilon/(4\beta)\right)\right)\right)$-UTT.

**Remark 1.** *The proof of Theorem 1, given in the Appendix, provides insight into how to find UTT with large $\rho$. The crucial assumption is that there exists $\boldsymbol{v}$ that can successfully adversarially attack the model $f^*(\boldsymbol{x})$ for sufficiently many 'bad' data. This is indeed verified in our experiments.*

### 3.3 ENFORCING $(\varepsilon, \delta)$-TROJAN TWIN MODEL VIA POISONING DATASET

Next, we describe how UTT induces TTMs. The crucial step is to poison a training set with UTT. In particular, given a UTT denoted as $\boldsymbol{v}$, let $g \in \mathcal{F}$ be the empirically twin model described in Definition 2, one can create the set of poisoned data

$$P_m := \{(\boldsymbol{x}_i + \boldsymbol{v}, 1 - y_i) \mid (\boldsymbol{x}_i, y_i) \in S_n, |1 - g(\boldsymbol{x}_i + \boldsymbol{v}) - g(\boldsymbol{x}_i)| \le \varepsilon\}.$$

We call $P_m$ the *$\boldsymbol{v}$-perturbed set*, and $m \ge \rho n$ is the size of $P_m$. $P_m$ can be mixed into the clean training set $S_n$ and their union is the *poisoned training set*. In the remainder of this section we will show that under mild assumptions, any hypothesis $f \in \mathcal{F}$ trained using the poisoned set that achieves low risk will be an $(\varepsilon, \delta)$-Trojan Twin model for some $\varepsilon$ and $\delta$. First, in Proposition 1 we show that the model trained using $P_m \cup S_n$ will behave like a TTM on the training dataset. In Theorem 2, we will generalize the guarantee on training set to distributional guarantee using some empirical analysis tools from (Pollard, 2012), concluding in Corollary 1 with the existence of a Trojaned Twin model with the distributional guarantee in Definition 1.

**Proposition 1** (Existence of Empirical TTM). *Assume we are given the training set $S_n$, a $(\xi, \varepsilon, \rho)$-UTT $\boldsymbol{v}$, the empirical twin model $g$, the $\boldsymbol{v}$-poisoned set $P_m$, and the empirical clean model $\widehat{f}$. Assume the interpolation condition: $\forall (\boldsymbol{x}_i, y_i) \in S_n$, $|\widehat{f}(x_i) - y_i| \le \varepsilon$. Let $\widetilde{f}$ be any hypothesis satisfying*
$$\sum_{(\boldsymbol{x}, y) \in S_n \bigcup P_m} (y - \widetilde{f}(\boldsymbol{x}))^2 \le \min_{f \in \mathcal{F}} \sum_{(\boldsymbol{x}, y) \in S_n \bigcup P_m} (y - f(\boldsymbol{x}))^2 + \varepsilon. \text{ We have:}$$

$$\frac{1}{n} \sum_{(\boldsymbol{x}_i, y_i) \in S_n} (y_i - \widetilde{f}(\boldsymbol{x}_i))^2 \le 4\varepsilon \tag{3}$$

$$\sum_{(\boldsymbol{x}_j + \boldsymbol{v}, 1 - y_j) \in P_m} \mathbb{1}\{(1 - y_j - \widetilde{f}(\boldsymbol{x}_j + \boldsymbol{v}))^2 \le \frac{8\varepsilon}{\rho}\} \ge \frac{m}{2} \tag{4}$$

Proposition 1 analyzes the consequence of training models on the poisoned dataset. The model that achieves low risk tends to fit both the clean and poisoned subset on the training set, which implies the succesfulness of finding TTM on the training set. In next Theorem, we generalize the results on training set to achieve a distributional guarantee. We analyze the 'in distribution' guarantees leveraging on standard tools for empirical process. The 'out of distribution' generalization guarantees, i.e., the generalization on the perturbed data set, require additional effort. We require Lipschitzness of hypothesis class and some regularity conditions of data distributions so that the empirical performance on a subset of training data will generalize.

**Theorem 2** (Existence of Distributional TTM). *Assume the hypothesis and conclusion of Proposition 1 hold, and $|y - f^*(\boldsymbol{x})| \le \varepsilon$ for all $(\boldsymbol{x}, y)$. Then, if $n \gtrsim \frac{1}{\varepsilon^4}\left(d_P(\mathcal{F})\log(\frac{1}{\varepsilon}) + \log(\frac{1}{\eta})\right)$, we have that $\mathbb{E}_{\boldsymbol{x}}[(\widetilde{f}(\boldsymbol{x}) - f^*(\boldsymbol{x}))^2] \le \varepsilon$ with probability at least $1 - \eta$.*

*In addition, assume all functions in $\mathcal{F}$ are $\beta$-Lipschitz, and that the density $\mu(\boldsymbol{x})$ is absolutely continuous with bounded and convex open support $\mathcal{D}$. Let $\tau = \inf_{\boldsymbol{x} \in \mathcal{D}} \mu\left(\mathcal{B}\left(\boldsymbol{x}, \varepsilon/(4\rho\beta)\right) \cap \mathcal{D}\right)$. For a set $\Omega \subset \mathcal{D}$, denote by $\mathcal{P}(\Omega, \varepsilon)$ an $\varepsilon$-sparse set of points in $\Omega$. Let $\mathcal{P}^*_{P_m} = \operatorname*{arg\,min}_{Q \subseteq P_m, |Q| \ge \frac{m}{2}} \max_{\mathcal{P}} |\mathcal{P}(Q, \varepsilon/(4\rho\beta))|$.*

*There exists $\Omega \subset \mathcal{D}$ s.t., $\mu\left(\Omega\right) \ge |\mathcal{P}^*_{P_m}|\tau$ and $\forall \boldsymbol{x} \in \Omega$, $(\widetilde{f}(\boldsymbol{x} + \boldsymbol{v}) - 1 + f^*(\boldsymbol{x}))^2 \le \frac{16\varepsilon}{\rho}$.*

Theorem 2 directly implies following Corollary:

**Corollary 1** (Existence of $(\varepsilon, \delta)$-TTM). *Given conditions in Theorem 2, $\widetilde{f}$ is a $(\varepsilon', \delta)$-Trojan Twin Model where $\varepsilon' \le \frac{16\varepsilon}{\rho}$ and $\delta \ge |\mathcal{P}^*_{P_m}|\tau$ with probability at least $1 - \eta$.*

**Remark 2.** *Theorem 2 implies some sufficient conditions for enforcing a $(\varepsilon, \delta)$-Trojan Twin Model, using the dataset poisoned by a UTT trigger. The lower bound of $\delta$ achieved in Corollary 1 is a*

*distributional dependent quantity and thus hard to derive a uniform bound. In practice, the attacker can train some models on the poisoned dataset and use a validation dataset to empirically evaluate value of δ. It can also be observed that a large value of ρ implies closer Trojan twin of f\* and wealthier data that can be manipulated, improving the quality of TTM.*

### 3.4 Algorithm in Practice: Generating the Universal Trojan Trigger

In this section we describe our algorithm for generating UTT. Our algorithm is well motivated by our theoretical analysis. The algorithm takes as input multiple clean models $\{f_1, ..., f_J\}$ where models are well trained and are of different variants. We introduce $J$ models to cover different hypothesis classes of classifiers. A discussion on the benefit of introducing multiple hypothesis classes can be found in Appendix Section A.5. The trigger flips data from the source class label $C_S$ to the target class label $C_T$. A target injection rate $\rho$ is given to control the fraction of data to be poisoned by $v$. The trigger has a budget $\xi$, $\|v\| \leq \xi$. Please see Appendix B.1 for more implementation details of our algorithm.

Our algorithm begins by sampling a perturbed data set $P_m = \{(x_1, C_S), \cdots, (x_m, C_S)\}$ conditional on the source class uniformly randomly. Here $m = \lceil \rho n \rceil$. During the iterative procedure, the algorithm computes a gradient direction of loss function $L^{(t)} = \sum_{j=1}^{J} \sum_{x \in P_m} l(C_T, f_j(x + v))$ to minimize the discrepancy between model output and target class on perturbed data $x + v$. In practice, we observe it suffices to pick $T$ to be some small constant, e.g., $T = 5$, to find satisfying $v$.

---

**Algorithm 1** `Universal Trojan Trigger Generation`

---

1: **Input:** Clean data set $S_n = \{(x_1, y_1), ..., (x_n, y_n)\} \subset R^d \times \{1, 2, \cdots, K\}$, pre-adversarial-trained clean model set $\{f_1, f_2, \cdots, f_J\}$, loss function $l$ (e.g., cross-entropy), randomly initialized Universal Trojan Trigger $v^{(0)} \in R^d$, source class $C_S$, target class $C_T$, trigger budget constraint $\xi$, learning rate $\eta$, injection fraction $\rho$, number of iterations $T$.

2: Sample perturbed set $P_m = \{(x_1, C_S), \cdots, (x_m, C_S)\}$ from label-$C_T$ data in $S_n$
3: **for** $t \leftarrow 1, \cdots, T$ **do**
4: $\quad L^{(t)} = \sum_{j=1}^{J} \sum_{x \in P_m} l(C_T, f_i(x + v^{(t-1)}))$
5: $\quad v^{(t)} = v^{(t-1)} - \eta \nabla_{v^{(t-1)}} L^{(t)}$
6: $\quad v^{(t)} = \xi v^{(t)} / \|v^{(t)}\|_2$
7: **end for**
8: **Output:** $v^{(T)}$

---

## 4 Experiment

In this section, we present and discuss the result of our empirical study. We first evaluate the attacking performance. We manually inject our UTT into different image datasets and use these datasets to train ResNet18 and VGG16 models. We then evaluate our method's performance against most recent backdoor attack baselines. We evaluate on various settings including the most challenging one where user adopts adversarial training. In the most challenging setting, all baselines has performance deterioration, whereas our method outperform others in this situation.

Next, we investigate the evasiveness of our method against Trojan detection methods. We Trojan multiple models with UTT and then investigate how resilient are these models against SOTA detection methods. Quantitative results show that our method is much more resilient to detection methods compared to other attacks. We also show our method's resistance to fine-pruning post-process. All these merits are implied by the properties of Trojan twin model.

Finally, we conduct ablation study on the choice of different hyper-parameters such as Trojan injection ratio and trigger size. Our method is shown to be robustness w.r.t. the choice of hyper-parameters.

### 4.1 Attack Experiments

**Experiment Setting.** In this section, we present the result of attacking experiments. We manually inject Trojan Trigger with each baseline attacking method into different dataset and the poisoned data

to ResNet18 (He et al., 2016) and VGG16 (Simonyan & Zisserman, 2014) for training. We fix the $L_2$ norm of each method's trigger to be 10 and the injection ratio to be 20% for each method. We train each method for 200 epochs with same batch size (128 for CIFAR10/GTSRB, 32 for IMAGENET), same learning rate 7e-3 (we use gradient accumulation due to the limited computation resource, so we scale original learning rate 1e-2 by $1/\sqrt{2}$ which is 7e-3) and same weight decay rate 5e-4. We will present ablation study results on injection ratio and trigger size in the appendix.

During the training of the model, we assume the most challenging situation where the user will adopt adversarial training (we use PGD here (Madry et al., 2017)). This test all baselines under a more practical setting because whenever the trigger is injected, we have no control over the training scheme that could be adopted by the user. For model-poisoning method like WaNet and adaptive attack method like IMC, we also use adversarial training to make fair comparison. For the reference, we present the result where we don't use adversarial training in the Appendix Table 4 - 5.

**Baselines.** We select several attacking methods that are representative for each schools mentioned in background section. We use name abbreviation BadNet for (Gu et al., 2017), SIG for (Barni et al., 2019), REF for (Liu et al., 2020), WaNet for (Nguyen & Tran, 2020) and IMC for (Pang et al., 2020). We have a detailed discussion of each baseline and the hyper-parameter setting of each baseline can be find in Appendix section B.1.

**Dataset.** We test our method against three image data set. CIFAR10 (Krizhevsky et al., 2009) is a small scale color dataset with 10 classes. Each image is of size 32×32. It has 50000 images for training and 10000 images for testing. GTSRB (Stallkamp et al., 2012) is the German traffic sign recognition dataset with 43 classes. We resize each image in GTSRB to 32×32. It has 26640 data points for training and 12630 data points for testing. We also test our method against ImageNet (Russakovsky et al., 2015). Because training on a large number of high resolution images results in training overhead. We pick images from class 0-9 from ILSVRC2012 dataset and resize each images from 224×224 to 112×112. It contains 13000 training images and 500 testing images.

**Evaluation Metrics.** As it suggested by our theoretical model, we should evaluate a attacking algorithm through two criterion. We conduct one-to-one attack here by choosing a source class and target class. One of the evaluation metric is to measure the attacking successful rate (ASR) on this source-target pair. ASR is the proportion of testing images from source class that could be mis-classified by the Trojaned models into the target class when edited by the trigger. The higher the ASR, the more effective the proposed attack.

Another evaluation metric is the classification accuracy (ACC), which measures the classification accuracy of a Trojaned model on clean images. We require a high ACC on a Trojaned model because we want the Trojaned model to keep intact functionality when it gets clean input.

**Discussion.** If we look at Appendix Tables 4-5, where no adversarial training is using during trainig of Trojaned model, all baseline achieve similar performance on both ACC and ASR for most of the case (we only highlight the significant best one using two sample t-tests). However, in practice database-poisoning methods (like BadNet, SIG, REF and ours) do not assume the access to the model. Thus these methods have no control over the training scheme adopted by users. For model-poisoning methods (like WaNet and IMC), their generated Trojaned models may be post-processed or fine tuned by the user. Users can adopt the training scheme that is the most unfavorable to attackers. Adversarial training is one of such training scheme that can hinder the Trojan attack. We can see from Table 1-2, even though all method suffer certain performance deterioration, our method maintain good ACC and consistently competitive ASR over all baselines. The advantage comes from the universal trigger generated by adversarial trained model pools. This specific trigger can manipulate the output of an adversarially trained model. User cannot avoid being Trojaned even they conduct adversarial training.

Table 1: Accuracy on Clean Inputs Under Adversarial Training

| Dataset | Network | BadNet | SIG | REF | WaNet | IMC | Ours |
|---------|---------|--------|-----|-----|-------|-----|------|
| CIFAR10 | ResNet18 | 0.902±0.003 | 0.912±0.003 | 0.905±0.002 | 0.901±0.005 | 0.909±0.001 | 0.908±0.002 |
|         | VGG16   | 0.897±0.002 | 0.903±0.001 | 0.902±0.001 | 0.900±0.002 | 0.900±0.000 | 0.904±0.001 |
| GTSRB   | ResNet18 | **0.925±0.003** | 0.910±0.013 | 0.904±0.019 | 0.911±0.011 | 0.899±0.004 | 0.912±0.002 |
|         | VGG16   | 0.941±0.002 | 0.944±0.006 | 0.942±0.002 | 0.938±0.001 | 0.939±0.004 | 0.946±0.009 |
| ImageNet | ResNet18 | 0.619±0.003 | 0.616±0.003 | 0.619±0.008 | 0.610±0.004 | 0.607±0.003 | 0.618±0.004 |
|         | VGG16   | 0.668±0.002 | 0.668±0.008 | 0.633±0.006 | 0.667±0.001 | 0.662±0.004 | 0.671±0.001 |

Table 2: Attack Successful Rate Under Adversarial Training

| Dataset | Network | BadNet | SIG | REF | WaNet | IMC | Ours |
|---------|---------|--------|-----|-----|-------|-----|------|
| CIFAR10 | ResNet18 | 0.992±0.001 | 0.957±0.016 | 0.746±0.002 | 0.966±0.009 | 0.988±0.002 | **0.994±0.000** |
|         | VGG16    | 0.990±0.003 | 0.957±0.002 | 0.731±0.004 | 0.960±0.007 | 0.978±0.003 | **0.994±0.000** |
| GTSRB   | ResNet18 | 0.969±0.007 | 0.904±0.083 | 0.885±0.033 | 0.950±0.019 | 0.892±0.030 | **0.978±0.000** |
|         | VGG16    | 0.973±0.003 | 0.956±0.014 | 0.881±0.028 | 0.926±0.047 | 0.569±0.071 | 0.976±0.004 |
| ImageNet | ResNet18 | 0.968±0.001 | 0.735±0.046 | 0.900±0.008 | 0.877±0.012 | 0.851±0.012 | 0.967±0.001 |
|          | VGG16    | 0.963±0.001 | 0.546±0.081 | 0.904±0.040 | 0.877±0.012 | 0.314±0.174 | **0.967±0.001** |

## 4.2 DETECTION TABLE

**Experiment Setting.** In this section we present the model inspection result. The number in Table 3 is copy from Table 11 of (Pang et al., 2022). We follow their experiment setting and Trojaned 10 ResNet18 models trained on CIFAR10 dataset. We use the same implementation of these model inspection algorithm and present the anomaly index value (AIV) number got by our method from each model investigation method on the last column of Table 3. We will discuss the evaluation metrics later in more detail.

**Baseline Attack.** There are mainly 8 attack methods compared in (Pang et al., 2020) including BadNet, REF and IMC, which we have discussed above. TNN is the method proposed in (Liu et al., 2017), TB is the blending method proposed in (Chen et al., 2017), LB is the method proposed in (Yao et al., 2019a), ABE is the method proposed in (Shokri et al., 2020). We have discussed all of these work in Appendix section B.1. Besides, embarrassingly simple backdoor attack (ESB) tries to attach an extra Trojaned neural net to the target model. They trained the merged network such that the Trojaned part will be activated whenever a trigger is presented otherwise the original part will activate. ESB belongs to model-poisoning attack method category. ABS doesn't apply to ESB simply because of its pre-requisite. If we assume a white box investigation, where investigator have access to the architecture, the capture of ESB is instant. If we assume black-box investigation, ABS is not applicable here.

**Baseline Defense.** We investigate all these attack methods with 5 widely used model-inspection algorithm. Neural cleanse (NC) (Wang et al., 2019), Deep Inspection (DI) (Chen et al., 2019), TABOR (Guo et al., 2019), Neuron Inspection (NI) (Chen et al., 2019) and Artificial Brain Stimulation (ABS) (Liu et al., 2019). More detailed discussion about these defense baselines can be found in Appendix section B.2.

**Evaluation Metric.** We mainly use anomaly index value (AIV) as the metric to recognize a Trojaned model. AIV is the normalized median absolute deviation. For a set of input $\{x_1, \cdots, x_n\}$, the median absolute deviation (MAD) is the median of $\{|x_1 - x_{\text{median}}|, \cdots, |x_n - x_{\text{median}}|\}$. Then the AIV of this set of points is $\{\frac{|x_1 - x_{\text{median}}|}{1.4826\text{MAD}}, \cdots, \frac{|x_n - x_{\text{median}}|}{1.4826\text{MAD}}\}$. Any point in this set of data that has an AIV larger than 2 is considered to be an outlier.

In our case, $n$ is the number of output classes and $x$ is the L1 norm of reversed trigger or explanatory feature (for ESB) given by each investigator. Following the setting of (Pang et al., 2022), for each output neuron in these 10 Trojaned networks, we record such AIV given by the target class. Then we perform a t-test for each attack-defense pair to decide if the AIV is significantly larger than 2 (MAD test). In the table 3, we highlight methods that are not evasive by corresponding investigation methods with †.

**Discussion.** From Table 3 we can see that most attacks are quite evasive against current detection algorithm. ABS is the most effective model-investigation method and capture TB, LB, ABE, and IMC. Our method is evasive to all listed investigation algorithm. This is partially suggested by our Theorem 2, which says the model Trojaned by our trigger represents a function that is very closed to what the clean model learnt. This can add difficulties to detection.

## 4.3 RESISTANCE AGAINST FINE PRUNING

In this section, we test our method's robustness against fine pruning post-processing. We implant Trojan into both RseNet18 and VGG16 models by poisoning the CIFAR10 dataset with our UTT. For

Table 3: AIV of Model-Inspection Method and Detection Algorithm. (Attack that are captured by corresponding inspection algorithm are highlighted with †)

| Defense \Attack | BadNet | TNN | REF | TB | LB | ESB | ABE | IMC | Ours |
|---|---|---|---|---|---|---|---|---|---|
| NC | 3.08 (±0.65) | 2.69 (±0.47) | 2.48 (±0.51) | 2.44 (±0.38) | 2.12 (±0.20) | 0.04 (±0.02) | 2.67 (±0.51) | 1.66 (±0.25) | 0.57 (±0.49) |
| DI | 0.54 (±0.06) | 0.46 (±0.04) | 0.39 (±0.04) | 0.29 (±0.03) | 0.21 (±0.04) | 0.01 (±0.00) | 0.76 (±0.10) | 0.26 (±0.03) | 2.12 (±1.31) |
| TABOR | 3.26 (±0.77) | 2.49 (±0.49) | 2.32 (±0.51) | 2.15 (±0.29) | 2.01 (±0.63) | 0.89 (±0.04) | $2.44^{\dagger}$ (±0.22) | 1.89 (±0.19) | 0.72 (±1.01) |
| NI | 1.28 (±0.21) | 0.59 (±0.11) | 0.78 (±0.06) | 1.11 (±0.34) | 0.86 (±0.87) | 0.71 (±0.10) | 0.41 (±0.05) | 0.52 (±0.13) | 0.87 (±1.45) |
| ABS | 3.02 (±0.81) | 4.16 (±1.33) | 4.10 (±1.27) | $15.55^{\dagger}$ (±6.59) | $2.88^{\dagger}$ (±0.25) | — | $8.45^{\dagger}$ (±3.22) | $3.15^{\dagger}$ (±0.43) | 3.31 (±4.07) |

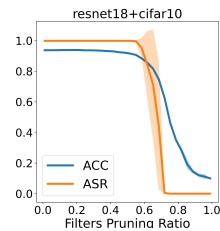
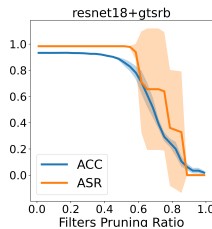
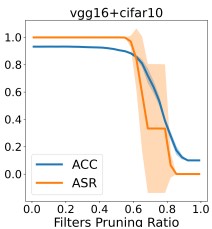
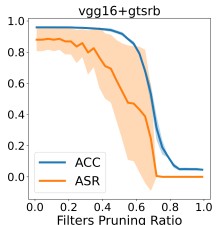

Figure 2: An illustration of the resistance of our method against find-pruning. The filter pruning ratio is the proportion of number of pruned filters over the total number of filters.

each convolutional layer we prune the convolutional filter with the lowest $L_1$ norm within current un-pruned filters in a stratified manner. We set different pruning ratios and measure ACC and ASR of the pruned network on the Trojan dataset. The result is shown in Figure 2. We can see that, the fine pruning cannot effectively reduce the ASR without hurting ACC. This is because the Trojan twin model is very closed to the clean model. It should not use too much extra capacity for recognizing the trigger. As a result, fine-pruning cannot easily erase the circuit for classifying trigger without damaging useful structure.

### 4.4 ABLATION STUDY ON INJECTION RATIO AND TRIGGER SIZE

In this section, we conduct ablation study on the effect of injection ratio and trigger size. In section 4.1, we fix the injection ratio to 20% and the $L_2$ norm of the trigger size to 10. In this section, we conduct an ablation study by reducing the injection ratio to 10% and reducing the trigger size to 5 separately, and test each baseline accordingly. Experiments results are presented in Appendix Tables 6-9. We can see that in the case where smaller trigger or fewer Trojan data is used, our method still maintains advantageous performance. These results corroborate our method's robustness against hyper-parameter choice.

## 5 SUMMARY

In this work, we study the Trojan Attack problem. We formulate the Trojan Attack task as finding a twin of clean model. We quantify the quality of twin model using a natural bi-criteria metric. We propose a poisoning data attacking strategy where the data is corrupted by our carefully designed trigger named UTT. We show the merit of our Trojan attack strategy both theoretically and empirically. In particular, empirical study shows that our method achieves competitive attacking effectiveness and detection resistance.

## 6 REPRODUCIBILITY STATEMENT

Our experiment uses only public available dataset. We have described our experiment setting and implementation details in Section 4. The source code will be made available together with the publication of this paper.

## 7 ETHICS STATEMENT

This paper studies the Trojan attack problem. Our study deepens understanding of Trojan attack and takes one step toward effective methods to defend against attack. The theory/method discussed in this work may be applied by malicious attackers to designed Trojan attack method that may cause security issues for DNN users.

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

## A  Theoretical Results Proof

### A.1  Preliminaries

Below we introducing some definitions that is used in our proof. The definitions of covering number, VC-dimensions and Pseudo-Dimensions can be found in (Pollard, 2012; Wellner et al., 2013; Mohri et al., 2018).

**Definition 3** ($L_2$-Covering Number). *Let $\boldsymbol{x}_{1:n}$ be set of points. A set of $U \subseteq \mathbb{R}^n$ is an $\varepsilon$-cover w.r.t $L_2$-norm of $\mathcal{F}$ on $x_{1:n}$, if $\forall f \in \mathcal{F}$, $\exists u \in U$, s.t. $\sqrt{\frac{1}{n} \sum_{i=1}^{n} |[u]_i - f(x_i)|^2} \le \varepsilon$, where $[u]_i$ is the $i$-th coordinate of $u$. The covering number $\mathcal{N}_2(\varepsilon, \mathcal{F}, n)$ with 2-norm of size $n$ on $\mathcal{F}$ is :*

$$\sup_{\boldsymbol{x}_{1:n} \in \mathcal{X}^n} \min\{|U|\text{: } U \text{ is an } \varepsilon\text{-cover of } \mathcal{F} \text{ on } x_{1:n}\} \tag{5}$$

**Definition 4** ($\beta$-Lipschitz). *We say hypothesis class $\mathcal{F}$ is $\beta$-Lipschitz if for all $f \in \mathcal{F}$ we have :*

$$|f(\boldsymbol{x}_1) - f(\boldsymbol{x}_2)| \le \beta \|\boldsymbol{x}_1 - \boldsymbol{x}_2\|$$

**Definition 5** (VC-dimension). *The VC-dimension $d_{VC}(\mathcal{F})$ of a hypothesis class $\mathcal{F} = \{f : \mathcal{X} \to \{1, -1\}\}$ is the largest cardinality of the a set $S \subseteq \mathcal{X}$ such that $\forall \bar{S} \subseteq S$, $\exists f \in \mathcal{F}$:*

$$f(x) = \begin{cases} 1 & \text{if } x \in \bar{S} \\ -1 & \text{if } x \in S \setminus \bar{S} \end{cases} \tag{6}$$

**Definition 6** (Pseudo-dimension). *The Pseudo-dimension $d_P(\mathcal{F})$ of a real-valued hypothesis class $\mathcal{F} = \{f : \mathcal{X} \to [a, b]\}$ is the VC-dimension of the hypothesis class $\mathcal{H} = \{h : \mathcal{X} \times \mathbb{R} \to \{-1, 1\} | h(\boldsymbol{x}, t) = sign(f(\boldsymbol{x}) - t), f \in \mathcal{F}\}$.*

**Definition 7** ($\varepsilon$-sparse set). *Given set of points $B \subseteq \mathbb{R}^d$ with finite size, we say $A$ is an $\varepsilon$-sparse set of $B$ if $A \subseteq B$ and $\forall a_1, a_2 \in A$, $\|a_1 - a_2\| \ge \varepsilon$.*

### A.2  Missing Proof for Theorem 1

**Proof**: For data For simplicity we denote $\|f_1 - f_2\|_{S_n} = \sqrt{\frac{1}{n} \sum_{i=1}^{n} (f_1(\boldsymbol{x}_i) - f_2(\boldsymbol{x}_i))^2}$ and $\|f_1 - f_2\|_{\mu(\boldsymbol{x})} = \sqrt{\mathbb{E}_{\boldsymbol{x}}[(f_1(\boldsymbol{x}) - f_2(\boldsymbol{x}))^2]}$. In the proof we denote $\gamma = \frac{\varepsilon}{4}$, $\rho = \frac{\mu\left(\mathcal{B}(\boldsymbol{x}_{\text{bad}}, \varepsilon/(4\beta))\right)}{2}$. Let $\mathcal{C}$ be a $\gamma^2$-cover for hypothesis class $\mathcal{F}$ projected on dataset with size $n$, for any hypothesis $f$ we denote $c(f)$ be element in $\mathcal{C}$ that covers $f$. In particular we have $\forall f, \exists c(f) \in T, \|c(f) - f\|_{S_n} \le \gamma^2$. Let $\widehat{f} = \arg\min_{f \in \mathcal{F}} \frac{1}{n} \sum_{i=1}^{n} (f(x_i) - y)^2$. It is easy to verify that

$$\frac{1}{n} \sum_{i=1}^{n} (c(\widehat{f})(\boldsymbol{x}_i) - y_i)^2$$

$$\le \frac{1}{n} \sum_{i=1}^{n} (c(\widehat{f})(\boldsymbol{x}_i) - \widehat{f}(\boldsymbol{x}_i) + \widehat{f}(\boldsymbol{x}_i) - y_i)^2$$

$$= \frac{1}{n} \sum_{i=1}^{n} (c(\widehat{f})(\boldsymbol{x}_i) - \widehat{f}(\boldsymbol{x}_i))^2 + (\widehat{f}(\boldsymbol{x}_i) - y_i)^2 + 2(c(\widehat{f})(\boldsymbol{x}_i) - \widehat{f}(\boldsymbol{x}_i))(\widehat{f}(\boldsymbol{x}_i) - y_i)$$

$$\le \left\{ \frac{1}{n} \sum_{i=1}^{n} (c(\widehat{f})(\boldsymbol{x}_i) - \widehat{f}(\boldsymbol{x}_i))^2 + (\widehat{f}(\boldsymbol{x}_i) - y_i)^2 \right\} + 2\sqrt{\frac{1}{n} \sum_{i=1}^{n} (c(\widehat{f})(\boldsymbol{x}_i) - \widehat{f}(\boldsymbol{x}_i))^2} \sqrt{\frac{1}{n} \sum_{i=1}^{n} (\widehat{f}(\boldsymbol{x}_i) - y_i)^2}$$

$$\tag{7}$$

which implies that

$$\frac{1}{n}\sum_{i=1}^{n}(c(\widehat{f})(\boldsymbol{x}_i)-y_i)^2 \leq \frac{1}{n}\sum_{i=1}^{n}(\widehat{f}(\boldsymbol{x}_i)-y_i)^2+\gamma^4+4\gamma^2 \leq \frac{1}{n}\sum_{i=1}^{n}(f^*(\boldsymbol{x}_i)-y_i)^2+\gamma^4+4\gamma^2.$$

By a standard empirical process argument, using Hoeffding type inequality with symmetricity (Pollard, 2012) and taking union bound on the covering set $T$, we have

$$\mathbb{P}\left[\sup_{f\in T}\left\{\left|\mathbb{E}_{\boldsymbol{x},y}\left[(f(\boldsymbol{x})-y)^2\right]-\frac{1}{n}\sum_{i=1}^{n}(f(\boldsymbol{x}_i)-y_i)^2\right|\geq\gamma^2\right\}\right] \leq 2\mathbb{E}_{S_n}[|T|]exp\left(-\frac{n\gamma^4}{2}\right)$$

we know that by picking $n\gtrsim\frac{\log(\frac{|T|}{\eta})}{\gamma^4}$

$$\mathbb{E}_{\boldsymbol{x},y}[(c(\widehat{f})(\boldsymbol{x})-y)^2]\leq\mathbb{E}_{\boldsymbol{x},y}[f^*(\boldsymbol{x})-y)^2]+9\gamma^2$$

together with the fact that $\mathbb{E}[y|\boldsymbol{x}]=f^*(\boldsymbol{x})$ implies that

$$\mathbb{E}_{\boldsymbol{x}}[(c(\widehat{f})(\boldsymbol{x})-f^*(\boldsymbol{x}))^2]\leq 9\gamma^2.$$

Next we show that

$$\frac{1}{n}\sum_{i=1}^{n}(c(\widehat{f})(\boldsymbol{x}_i)-f^*(\boldsymbol{x}_i))^2\leq 13\gamma^2.$$

We apply Hoeffding type inequality to show that the probability of following event, if we draw $S_n$ in an i.i.d fashion with $n\gtrsim\frac{1}{\gamma^4}\log\left(\frac{\mathbb{E}_{S_n}[|T|]}{\delta}\right)$, is at least $1-\eta$:

$$\forall c(f)\in T, \left|\frac{1}{n}\sum_{i=1}^{n}(c(f)(\boldsymbol{x}_i)-f^*(\boldsymbol{x}_i))^2-\mathbb{E}_{\boldsymbol{x}}[c(f)(\boldsymbol{x})-f^*(\boldsymbol{x})^2]\right|\leq 13\gamma^2 \tag{8}$$

To see this, we invoke Hoeffding type inequality again (Pollard, 2012)

$$\mathbb{P}_{S_n}\left[\sup_{f\in\mathcal{F}}\left|\frac{1}{n}\sum_{i=1}^{n}(c(\widehat{f})(\boldsymbol{x}_i)-f^*(\boldsymbol{x}_i))^2-\mathbb{E}_{\boldsymbol{x}}[c(\widehat{f})(\boldsymbol{x})-f^*(\boldsymbol{x})^2]\right|\geq\gamma^2\right]\leq 2\mathbb{E}_{S_n}[|T|]exp\left(-\frac{\gamma^4 n}{4}\right) \tag{9}$$

We can show $n\gtrsim\frac{1}{\gamma^4}\log\left(\frac{\mathbb{E}_{S_n}[|T|]}{\eta}\right)$ implies that Inequality 8 holds with probability at least $1-\eta$.

Next we bound $|T|$. By Theorem 2.6.4 in (Wellner et al., 2013) we know that there exists universal constants $K,C<\infty$, for all $\boldsymbol{x}_{1:n}$, $|T|\leq Cd_P(\mathcal{F})K^{d_P(\mathcal{F})}\left(\frac{1}{\varepsilon}\right)^{2d_P(\mathcal{F})}$, which implies that it suffices to pick $n\gtrsim\frac{d_P(\mathcal{F})\log(\frac{1}{\gamma})+\log(\frac{1}{\eta})}{\gamma^4}$ to ensure that 8 holds with probability at least $1-\eta$.

Now we can bound the difference between $\widehat{f}$ and $f^*$ on under empirical $L$-1 metric:

$$\frac{1}{n}\sum_{i=1}^{n}|f^*(\boldsymbol{x}_i)-\widehat{f}(\boldsymbol{x}_i)|$$

$$\leq\sqrt{\frac{1}{n}}\sqrt{\sum_{i=1}^{n}(f^*(\boldsymbol{x}_i)-\widehat{f}(\boldsymbol{x}_i))^2}=\|f^*-\widehat{f}\|_{S_n} \tag{10}$$

$$\leq\|f^*-c(\widehat{f})\|_{S_n}+\|\widehat{f}-c(\widehat{f})\|_{S_n}$$
$$\leq 4\gamma$$

Let $S_{\text{bad}} = S_n \cap \mathcal{B}(\boldsymbol{x}_{\text{bad}}, \varepsilon/(4\beta))$. The choice of $n$ also implies that $|S_{\text{bad}}| \geq \frac{\rho}{2}$ with probability at least $1 - \eta$. For any $\boldsymbol{x}' \in S_{\text{bad}}$, we have:

$$
\begin{aligned}
&|1 - f^*(\boldsymbol{x}' + \boldsymbol{v}) - f^*(\boldsymbol{x}')| \\
\leq &|1 - f^*(\boldsymbol{x}_{\text{bad}} + \boldsymbol{v}) + f^*(\boldsymbol{x}_{\text{bad}} + \boldsymbol{v}) - f^*(\boldsymbol{x}' + \boldsymbol{v}) - f^*(\boldsymbol{x}_{\text{bad}}) + f^*(\boldsymbol{x}_{\text{bad}}) - f^*(\boldsymbol{x}')| \\
\leq &|1 - f^*(\boldsymbol{x}_{\text{bad}} + \boldsymbol{v}) - f^*(\boldsymbol{x}_{\text{bad}})| + |f^*(\boldsymbol{x}_{\text{bad}} + \boldsymbol{v}) - f^*(\boldsymbol{x}' + \boldsymbol{v})| + |f^*(\boldsymbol{x}_{\text{bad}}) - f^*(\boldsymbol{x}')| \quad (11) \\
\leq &\varepsilon + \beta\frac{\varepsilon}{4\beta} + \beta\frac{\varepsilon}{4\beta}
\end{aligned}
$$

Thus, with probability at least $1 - 3\delta$, we have existence of $f \in \mathcal{F}$ and $\boldsymbol{v}$ s.t.,

$$
\begin{aligned}
&\frac{1}{n}\sum_{i=1}^{n}|f(\boldsymbol{x}_i) - \widehat{f}(\boldsymbol{x}_i)| \leq \varepsilon \\
&\frac{1}{n}\sum_{i=1}^{n}\mathbb{1}\{|f(\boldsymbol{x}_i + \boldsymbol{v}) - 1 + f(\boldsymbol{x}_i)| \leq 2\varepsilon\} \geq \frac{\rho}{2} \\
&\|\boldsymbol{v}\| \leq \xi
\end{aligned}
$$

(12)

$\square$

**Remark 3.** *The crucial assumption in proving Theorem 1 is that there exists $\boldsymbol{v}$ that can successfully adversarial attack the model $f^*(\boldsymbol{x})$ for at least one 'bad' data. In practice, several observations have been made that there exists some common direction $\boldsymbol{v}$ which can adversarially attack many $\boldsymbol{x}_{\text{bad}}$ that are distinct from each other (Moosavi-Dezfooli et al., 2017). In such scenario, $\rho$ could be significantly increased if the common direction is found.*

### A.3 Missing Proof for Proposition 1

**Proof**: On one hand we know

$$
\begin{aligned}
&\frac{1}{n}\sum_{(\boldsymbol{x}_i,y_i)\in S_n}(y_i - g(\boldsymbol{x}_i))^2 \leq \frac{1}{n}\sum_{(\boldsymbol{x}_i,y_i)\in S_n}|y_i - g(\boldsymbol{x}_i)| \\
\leq &\frac{1}{n}\sum_{(\boldsymbol{x}_i,y_i)\in S_n}|y_i - \widehat{f}(\boldsymbol{x}_i) + \widehat{f}(\boldsymbol{x}_i) - g(\boldsymbol{x}_i)| \\
\leq &\frac{1}{n}\sum_{(\boldsymbol{x}_i,y_i)\in S_n}\left\{|y_i - \widehat{f}(\boldsymbol{x}_i)| + |\widehat{f}(\boldsymbol{x}_i) - g(\boldsymbol{x}_i)|\right\} \\
\leq &2\varepsilon.
\end{aligned}
$$

(13)

We also have:

$$
\begin{aligned}
&\frac{1}{m}\sum_{(\boldsymbol{x}_j+\boldsymbol{v},1-y_j)}(g(\boldsymbol{x}_j + \boldsymbol{v}) - 1 + y_j)^2 \\
\leq &\frac{1}{m}\sum_{(\boldsymbol{x}_j+\boldsymbol{v},1-y_j)}|g(\boldsymbol{x}_j + \boldsymbol{v}) - 1 + y_j| \\
\leq &\frac{1}{m}\sum_{(\boldsymbol{x}_j+\boldsymbol{v},1-y_j)}|g(\boldsymbol{x}_j + \boldsymbol{v}) - 1 + g(\boldsymbol{x}_j) - g(\boldsymbol{x}_j) + y_j| \\
\leq &\frac{1}{m}\sum_{(\boldsymbol{x}_j+\boldsymbol{v},1-y_j)}|g(\boldsymbol{x}_j + \boldsymbol{v}) - 1 - g(\boldsymbol{x}_j)| + \frac{1}{m}\sum_{(\boldsymbol{x}_j,y_j)}|g(\boldsymbol{x}_j) - \widehat{f}(\boldsymbol{x}_j) + \widehat{f}(\boldsymbol{x}_j) - y_j| \\
\leq &2\varepsilon + \frac{2n}{m}\varepsilon
\end{aligned}
$$

(14)

Thus we have

$$
\frac{1}{m+n} \sum_{(\boldsymbol{x}_i, y_i) \in S_n \bigcup P_m} (y_i - \widetilde{f}(\boldsymbol{x}_i))^2
$$

$$
\leq \frac{1}{m+n} \sum_{(\boldsymbol{x}_i, y_i) \in S_n \bigcup P_m} (y_i - \widetilde{g}(\boldsymbol{x}_i))^2 + \varepsilon \tag{15}
$$

$$
\leq \frac{n}{m+n} \left\{ \frac{1}{n} \sum_{(\boldsymbol{x}_i, y_i) \in S_n} (y_i - g(\boldsymbol{x}_i))^2 \right\} + \frac{m}{m+n} \left\{ \frac{1}{m} \sum_{(\boldsymbol{x}_j + \boldsymbol{v}, 1 - y_j)} (g(\boldsymbol{x}_j + \boldsymbol{v}) - 1 + y_j)^2 \right\}
$$

$$
\leq 3\varepsilon
$$

Since

$$
\sum_{(\boldsymbol{x}_i, y_i) \in S_n} (y_i - \widetilde{f}(\boldsymbol{x}_i))^2 \leq \sum_{(\boldsymbol{x}_i, y_i) \in S_n \bigcup P_m} (y_i - \widetilde{f}(\boldsymbol{x}_i))^2
$$

and

$$
\sum_{(\boldsymbol{x}_j + \boldsymbol{v}, 1 - y_j)} (\widetilde{f}(\boldsymbol{x}_j + \boldsymbol{v}) - 1 + y_j)^2 \leq \sum_{(\boldsymbol{x}_i, y_i) \in S_n \bigcup P_m} (y_i - \widetilde{f}(\boldsymbol{x}_i))^2
$$

Thus we conclude that the following two inequality holds:

$$
\frac{1}{n} \sum_{(\boldsymbol{x}_i, y_i) \in S_n} (y_i - \widetilde{f}(\boldsymbol{x}_i))^2 \leq 4\varepsilon \tag{16}
$$

$$
\frac{1}{m} \sum_{(\boldsymbol{x}_j + \boldsymbol{v}, 1 - y_j) \in P_m} (1 - y_j - \widetilde{f}(\boldsymbol{x}_j + \boldsymbol{v}))^2 \leq \frac{2(m+n)\varepsilon}{m} \leq \frac{4\varepsilon}{\rho} \tag{17}
$$

By Markov inequality, at least $\frac{m}{2}$ points satisfies

$$
(1 - y_j - \widetilde{f}(\boldsymbol{x}_j + \boldsymbol{v}))^2 \leq \frac{8\varepsilon}{\rho} \tag{18}
$$

$\square$

## A.4 Missing Proof for Theorem 2

**Proof**: An empirical process argument similar to the proof in Theorem 1 show that as long as $n \gtrsim \frac{d_P(\mathcal{F}) \log(\frac{1}{\varepsilon}) + \log(\frac{1}{\eta})}{\varepsilon^4}$, we have: $\mathbb{E}_{\boldsymbol{x}}[(y - \widetilde{f}(\boldsymbol{x}))^2] \lesssim \varepsilon$, which implies that $\mathbb{E}_{\boldsymbol{x}}[f^*(\boldsymbol{x}) - \widetilde{f}(\boldsymbol{x}))^2] \lesssim \varepsilon$. By Proposition 1, at least $\frac{m}{2}$ points satisfies :

$$
(1 - y_j - \widetilde{f}(\boldsymbol{x}_j + \boldsymbol{v}))^2 \leq \frac{8\varepsilon}{\rho} \tag{19}
$$

Next we construct $\Omega$. Let $Q$ be set of points in $P_m$ such that Equation 19 are satisfied. We set $\Omega = \bigcup_{\boldsymbol{x} \in Q} \mathcal{B}(\boldsymbol{x}, \frac{\varepsilon}{4\rho\beta}) \cap \mathcal{D}$. Let $\mathcal{P}^*(Q, \frac{\varepsilon}{4\beta\rho})$ be maximum $\frac{\varepsilon}{4\beta\rho}$-sparse set of $Q$, i.e., $\forall a, b \in Q, \|a - b\| \geq \frac{\varepsilon}{4\rho\beta}$. Since for arbitrary subset of $P_m$ with size at least $\frac{m}{2}$, its max $\frac{\varepsilon}{4\rho\beta}$-packing number is at least $|\mathcal{P}^*_{P_m}|$, thus $\eta = \mu(\Omega) \geq |\mathcal{P}^*_{P_m}|\tau$. For any $\boldsymbol{x}' \in \Omega$, we can find $\boldsymbol{x}_{\text{bad}} \in Q$ s.t. $\|\boldsymbol{x}_{\text{bad}} - \boldsymbol{x}'\| \leq \frac{\varepsilon}{4\rho\beta}$. Thus $|f^*(\boldsymbol{x}_{\text{bad}}) - f^*(\boldsymbol{x}')| \leq \frac{\varepsilon}{4\rho}$ and $|\widetilde{f}(\boldsymbol{x}_{\text{bad}}) - \widetilde{f}(\boldsymbol{x}')| \leq \frac{\varepsilon}{4\rho}$. Thus we have for all $\boldsymbol{x}' \in \Omega$:

$$
\begin{aligned}
&((\widetilde{f}(\boldsymbol{x}' + \boldsymbol{v}) - 1 + f^*(\boldsymbol{x}'))^2 \\
\leq& (\widetilde{f}(\boldsymbol{x}' + \boldsymbol{v}) - \widetilde{f}(\boldsymbol{x} + \boldsymbol{v}) + \widetilde{f}(\boldsymbol{x} + \boldsymbol{v}) - 1 + f^*(\boldsymbol{x}') - f^*(\boldsymbol{x}) + f^*(\boldsymbol{x}))^2 \\
\leq& 4|\widetilde{f}(\boldsymbol{x}' + \boldsymbol{v}) - \widetilde{f}(\boldsymbol{x} + \boldsymbol{v}) + \widetilde{f}(\boldsymbol{x} + \boldsymbol{v}) - 1 + f^*(\boldsymbol{x}') - f^*(\boldsymbol{x}) + f^*(\boldsymbol{x})|) \\
\leq& 4(\frac{\varepsilon}{4\rho\beta}\beta + \frac{\varepsilon}{4\rho\beta}\beta + |\widetilde{f}(\boldsymbol{x} + \boldsymbol{v}) - 1 + f^*(\boldsymbol{x})|) \\
\leq& \frac{22\varepsilon}{\rho}
\end{aligned} \tag{20}
$$

$\square$

A.5 DEFINITION AND DISCUSSION OF MULTI-HYPOTHESIS UTT

Corollary 1 suggests that the data poisoning schema using universal Trojan trigger is indeed very powerful in enforcing user's model to be a TTM . Under the assumption of Corollary 1, as long as user's model achieves low risk on the poisoned dataset, the model becomes a TTM with high probability. One important assumption made in Corollary 1 is that the choice of hypothesis by users, e.g., the architecture, belongs to the hypothesis class that attacker considered when poisoning the dataset. To ensure the assumption holds, we generalize the definition of UTT into following multi-hypothesis class version to cover more type of neural networks. Our Corollary 1 can be easily generalized to the following multi-hypothesis class version.

**Definition 8** (UTT for union of hypothesis classes). *Given data set $S_n = \{(\boldsymbol{x}_i, y_i)\}_{i=1}^n$ and family of hypothesis class $\mathcal{F} = \bigcup_{j=1}^J \mathcal{F}_j$, let $\widehat{f}_j(x) = \arg\min_{f \in \mathcal{F}_J} \sum_{i=1}^n (y_i - f(\boldsymbol{x}_i))^2$. We say $\boldsymbol{v}$ is $\boldsymbol{v}(\xi, \varepsilon, \rho)$ -UTT for hypothesis $\bigcup_{k=1}^j \mathcal{F}_j$ if for every hypothesis class $\mathcal{F}_j$ there exists some $f \in \mathcal{F}_j$ s.t.:*

$$\frac{1}{n}\sum_{i=1}^n |f(\boldsymbol{x}_i) - \widehat{f}_j(\boldsymbol{x}_i)| \le \varepsilon$$

$$\frac{1}{n}\sum_{i=1}^n \mathbb{1}\{|1 - f(\boldsymbol{x}_i + \boldsymbol{v}) - f(\boldsymbol{x}_i)| \le \varepsilon\} \ge \rho \qquad (21)$$

$$\|\boldsymbol{v}\| \le \xi$$

# B  EXPERIMENTS SETTING AND DETAILS

## B.1  ATTACK BASELINES.

**BadNet** (Gu et al., 2017) places a 3×3 image patches on the corner of a Trojan images as a trigger. Labels of Trojan images are also changed to the target class at the same time. Attacker inject these modified data point and create the Trojan database. **SIG** (Barni et al., 2019) overlay the target image with a watermark where each pixel has a sinusoidal function value depends on the pixel's position in the image, which makes the trigger invisible to human eyes. **REF** further improve the trigger stealthiness by using the reflection effect. They blend the trigger image into the target image and make the trigger looks like a natural reflection. **WaNet** Nguyen & Tran (2020) propose to use the warping operation to create trojan images. Instead of use a predefined trigger, they continue to apply warping operation on clean images during training and these warped images are considered trojan images and their labels are modified into target classes during training. **IMC** Pang et al. (2020) propose a bi-level optimization procedure that optimize the trigger and model at the same time to minimize the empirical loss on the trojan database. **TNN** Liu et al. (2017) is an adaptive attacke method that optimize to find the trigger that maximize the output of specific neuron in the penultimate layer. **TB** Chen et al. (2017) uses natural images as the water mark trigger and blend these trigger with source images to achieve visual stealthness. **ABE** Shokri et al. (2020) use GAN to generate trigger that produce indistinguishable intermediate layer representation as clean instance does. **LB** Yao et al. (2019a) follows the idea of BadNet but add restriction on the intermediate layer representation of Trojan images to be more closed to clean instances.

**Hyper-parameter Setting.** Baseline **BadNet** and **SIG** don't have specific hyper-parameters. For **REF**, we follow the original paper setting. We use PASCAL VOC dataset Everingham et al. (2010) as the candidate trigger base. We select trigger outof the whole PASCAL dataset. We finally keep 200 triggers given by the trigger search procedure. For **WaNet**, we also use the original setting, we set the hyper-parameter $K$ to be 4 and $S$ to be 0.5. We use cross-rate 2. For **IMC**, we use the default setting and conduct 1 iteration adversarial attack searching for the trigger and 1 iteration of model update iteratively using learning rate 0.1. For **Our** method, we use 5 adversarially trained models to search for the a single UTT. We adopt 5 step adversarial training to search for UTT.

**Implementation Details of Algorithm 1.** We use 5 adversarially-pretrained model to search for the universal Trojan trigger. Each model in the pool is initialized differently from those used in the attacking experiment. In Table 1-2, we use same architecture for the pools and for the target model. We also present the transferring attack result in Table 10 where we search the trigger with architecture A and attack model in architecture B. For example, we inject trigger found with VGG16 to attack ResNet18 model, we can get similar performance as the original paper.

## B.2  DEFENSE BASELINES.

**Neural Cleanse (NC)** is a trigger reversion method. It optimizes a randomly initialized pattern until the patter can change the output of the model under investigation. A model is recognized as Trojaned if the size of the reversed pattern is small. **Deep inspection (DI)** (Chen et al., 2019) use GAN, instead of trigger inversion, to generate trigger candidate in order to change the output of the target model. Then a model is detected to be Trojaned if the generated trigger's mask MAD goes beyond 2. **TABOR** (Guo et al., 2019) follows the idea of NC, but add regularization terms to enforce the reversed pattern to have similar shape and placement location as real trigger does. **Neuron inspection (NI)** (Huang et al., 2019) calculate several explanatory feature using the gradient heat map of the target model for Trojan detection. **Artificial brain stimulation (ABS)** (Liu et al., 2019) identify suspicious neurons in the target model by adding stimulus value to each neuron's output. A neuron is identified as compromised neurons if stimulation to this neuron can maximally change the output of the target network. Then a reverse engineering process is used to find a candidate trigger that can maximally stimulate the compromised neuron.

## C   MORE EXPERIMENTS RESULTS

Table 4: Accuracy on Clean Inputs without Adversarial Training

| Dataset | Network | BadNet | SIG | REF | WaNet | IMC | Ours |
|---|---|---|---|---|---|---|---|
| CIFAR10 | ResNet18 | 0.941±0.002 | 0.943±0.001 | 0.942±0.002 | 0.937±0.001 | 0.943±0.002 | 0.942±0.002 |
| | VGG16 | 0.932±0.001 | 0.934±0.001 | 0.933±0.001 | 0.930±0.001 | 0.934±0.001 | 0.935±0.000 |
| GTSRB | ResNet18 | 0.933±0.002 | 0.941±0.008 | 0.941±0.007 | 0.939±0.015 | 0.942±0.002 | 0.940±0.003 |
| | VGG16 | 0.945±0.002 | 0.959±0.001 | 0.957±0.005 | 0.960±0.002 | 0.959±0.003 | 0.957±0.003 |
| ImageNet | ResNet18 | 0.691±0.014 | 0.692±0.007 | 0.700±0.003 | 0.686±0.004 | 0.700±0.003 | 0.699±0.017 |
| | VGG16 | 0.786±0.009 | 0.792±0.005 | 0.785±0.008 | 0.785±0.009 | 0.785±0.008 | 0.781±0.008 |

Table 5: Attack Successful Rate without Adversarial Training

| Dataset | Network | BadNet | SIG | REF | WaNet | IMC | Ours |
|---|---|---|---|---|---|---|---|
| CIFAR10 | ResNet18 | 0.999±0.000 | 0.998±0.001 | 0.805±0.006 | 0.998±0.001 | 0.999±0.000 | 0.999±0.000 |
| | VGG16 | 0.999±0.000 | 0.997±0.001 | 0.810±0.008 | 0.997±0.001 | 0.999±0.000 | 0.999±0.000 |
| GTSRB | ResNet18 | 0.984±0.000 | 0.984±0.000 | 0.951±0.016 | 0.984±0.000 | 0.984±0.000 | 0.984±0.000 |
| | VGG16 | 0.984±0.000 | 0.984±0.000 | 0.973±0.019 | 0.967±0.028 | 0.978±0.009 | 0.984±0.000 |
| ImageNet | ResNet18 | 0.980±0.000 | 0.948±0.023 | 0.980±0.000 | 0.967±0.011 | 0.980±0.000 | 0.980±0.000 |
| | VGG16 | 0.980±0.000 | 0.824±0.079 | 0.974±0.011 | 0.974±0.011 | 0.974±0.011 | 0.980±0.000 |

Table 6: Ablation Study on Injection Ratio: ACC

| Dataset | Network | BadNet | SIG | REF | WaNet | IMC | Ours |
|---|---|---|---|---|---|---|---|
| CIFAR10 | ResNet18 | 0.903±0.003 | **0.912±0.003** | 0.905±0.002 | 0.905±0.001 | 0.909±0.002 | 0.908±0.002 |
| | VGG16 | 0.901±0.003 | 0.903±0.001 | 0.902±0.001 | 0.901±0.001 | 0.902±0.001 | 0.904±0.001 |
| GTSRB | ResNet18 | 0.900±0.016 | 0.910±0.013 | 0.904±0.019 | 0.911±0.008 | 0.901±0.011 | 0.899±0.007 |
| | VGG16 | 0.936±0.003 | 0.944±0.006 | 0.942±0.002 | 0.942±0.003 | 0.946±0.003 | 0.940±0.007 |

Table 7: Ablation Study on Injection Ratio: ASR

| Dataset | Network | BadNet | SIG | REF | WaNet | IMC | Ours |
|---|---|---|---|---|---|---|---|
| CIFAR10 | ResNet18 | 0.991±0.000 | 0.957±0.016 | 0.746±0.002 | 0.926±0.021 | 0.984±0.003 | **0.994±0.000** |
| | VGG16 | 0.992±0.001 | 0.957±0.002 | 0.731±0.004 | 0.914±0.016 | 0.976±0.001 | **0.994±0.000** |
| GTSRB | ResNet18 | 0.970±0.004 | 0.904±0.083 | 0.885±0.033 | 0.895±0.042 | 0.806±0.010 | **0.976±0.001** |
| | VGG16 | 0.971±0.009 | 0.956±0.014 | 0.881±0.028 | 0.938±0.019 | 0.341±0.110 | 0.975±0.003 |

Table 8: Ablation Study on Trigger Size: ACC

| Dataset | Network | BadNet | SIG | REF | WaNet | IMC | Ours |
|---|---|---|---|---|---|---|---|
| CIFAR10 | ResNet18 | 0.905±0.002 | 0.908±0.002 | 0.903±0.004 | 0.902±0.003 | 0.909±0.001 | 0.906±0.002 |
| | VGG16 | 0.900±0.002 | 0.901±0.001 | 0.900±0.001 | 0.900±0.002 | 0.900±0.000 | 0.902±0.003 |
| GTSRB | ResNet18 | 0.899±0.002 | 0.908±0.021 | 0.913±0.012 | 0.910±0.008 | 0.899±0.004 | 0.914±0.002 |
| | VGG16 | 0.937±0.003 | 0.942±0.006 | 0.934±0.004 | 0.940±0.001 | 0.939±0.004 | 0.938±0.007 |

Table 9: Ablation Study on Trigger Size: ASR

| Dataset | Network | BadNet | SIG | REF | WaNet | IMC | Ours |
|---|---|---|---|---|---|---|---|
| CIFAR10 | ResNet18 | 0.990±0.001 | 0.866±0.029 | 0.757±0.007 | 0.972±0.002 | 0.988±0.002 | 0.991±0.004 |
| | VGG16 | 0.989±0.001 | 0.888±0.013 | 0.755±0.011 | 0.969±0.003 | 0.978±0.003 | **0.994±0.000** |
| GTSRB | ResNet18 | 0.967±0.002 | 0.826±0.032 | 0.859±0.024 | 0.945±0.020 | 0.892±0.030 | **0.975±0.003** |
| | VGG16 | 0.968±0.006 | 0.734±0.088 | 0.849±0.041 | 0.936±0.016 | 0.569±0.071 | **0.977±0.001** |

Table 10: Transferring Attack Result of Our Method

| Dataset | Pooled Network | Target Network | ACC | ASR |
|---|---|---|---|---|
| CIFAR10 | VGG16 | ResNet18 | 0.907±0.004 | 0.994±0.001 |
| CIFAR10 | ResNet18 | VGG16 | 0.905±0.001 | 0.994±0.000 |
| GTSRB | VGG16 | ResNet18 | 0.914±0.050 | 0.976±0.001 |
| GTSRB | ResNet18 | VGG16 | 0.939±0.004 | 0.977±0.001 |

**Global Pruning Results.** We also provide the global pruning result, where we prune the filter that has the smallest L1 norm among all convolutional layer instead of doing it in a stratified manner. With this pruning method, it is possible some layers can be totally removed during the increasing of pruning ratio. We present the added experiments in appendix Figure 3. We could also observe similar resistance result as it is in layer-wise pruning.

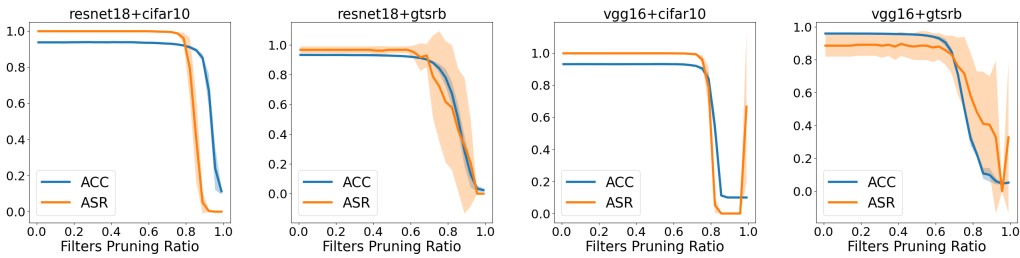

Figure 3: An illustration of the resistance of our method against global find-pruning. The filter pruning ratio is the proportion of number of pruned filters over the total number of filters.

## D    DEMONSTRATION OF TROJAN IMAGES AND TRIGGERS

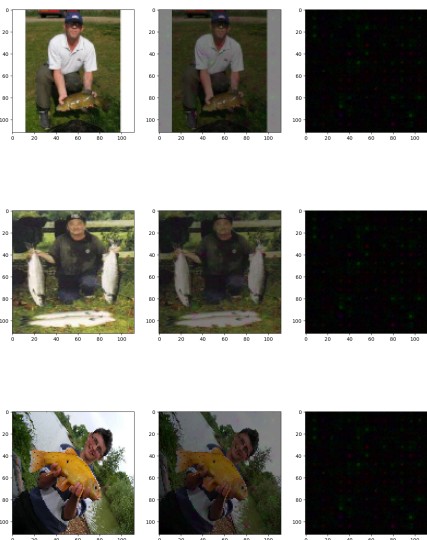

Figure 4: An illustration of UTT triggers on ImageNet. Left column displays the clean images. Middle column displays the Trojan images. Right column displays the Trojan triggers. We increased the transparency for middle column when overlay triggers, so it looks darker.

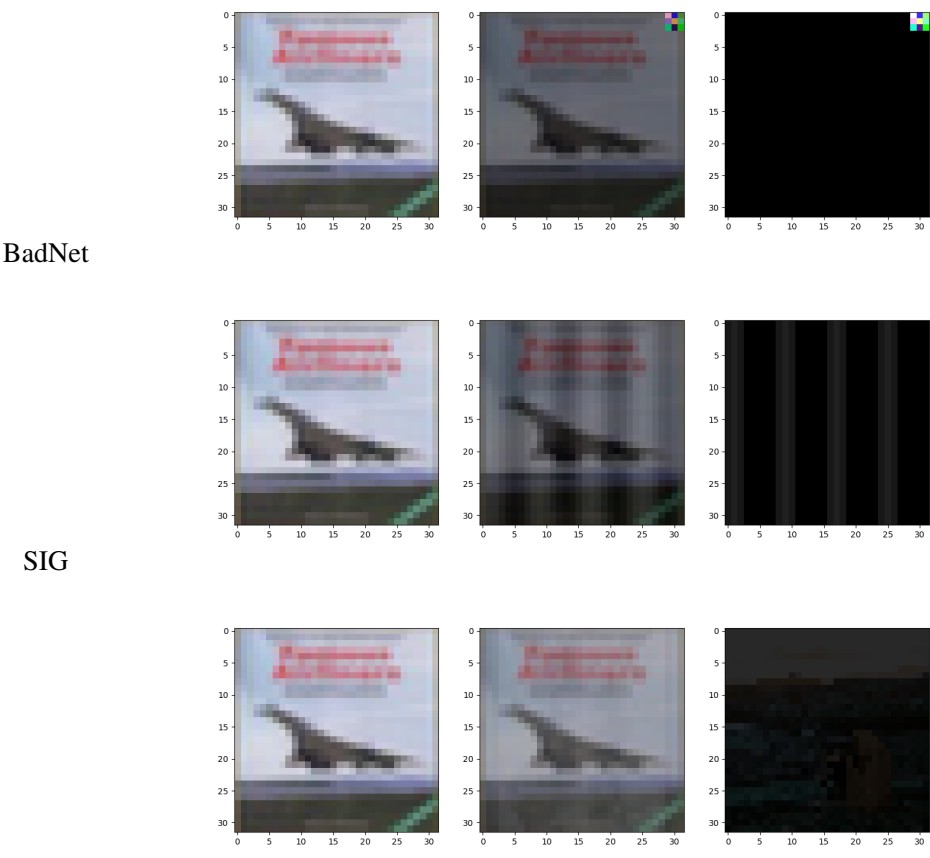

BadNet

SIG

REF

Figure 5: An illustration of triggers found by each baselines with CIFAR10. Left column displays the clean images. Middle column displays the Trojan images. Right column displays the Trojan triggers.

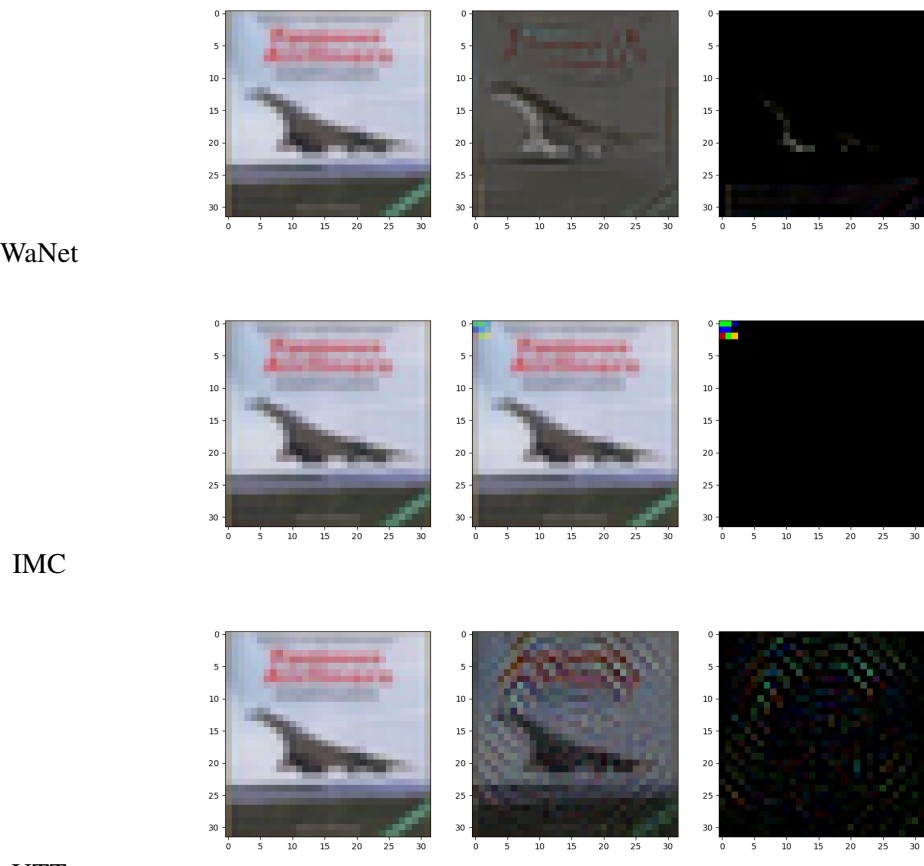

WaNet

IMC

UTT

Figure 6: Continued - An illustration of triggers found by each baselines with CIFAR10. Left column displays the clean images. Middle column displays the Trojan images. Right column displays the Trojan triggers.

