# OpenReview forum: "On the Existence of a Trojaned Twin Model"
_ICLR.cc/2023/Conference — Submitted to ICLR 2023_

### Official Review · Reviewer_YzEQ · 2022-10-17

**Confidence:** 4
**Correctness:** 3
**Technical Novelty And Significance:** 3
**Empirical Novelty And Significance:** 3
**Recommendation:** 6

**Clarity, Quality, Novelty And Reproducibility:**

The clarity and quality of this paper are good, and I admire the novelty of the idea of Trojaned Twin Model. The reproducibility is not check because no code is provided.

**Details Of Ethics Concerns:**

This paper proposes a universal poisoning attack, which may result in some ethic problem.

**Strength And Weaknesses:**

Strength:
- The idea of Trojaned Twin Model is interesting, and Definition 2 makes sense to me.

- The writing is clear and the formulation used in this paper is in the appropriate format.

- The empirical evaluation is done on different datasets, while several baselines are compared.

Weaknesses:
- As to Theorem 1, there is $|1-f^*(x_{bad}+v)-f^*(x_{bad})|<\epsilon$. For a practical distribution $\mu(x)$, if $\epsilon$ is small (satisfying the definition of Trojan Twin), then $x_{bad}$ is usually located at low-probability regions. So consequently, $\mu(\mathcal{B}(x_{bad},\epsilon/(4\beta)))$, i.e., the ratio of successful trojan $\rho$ would be quite small. Besides, practical model family $\mathcal{F}$ usually has a large value of Lipschitz constant $\beta$, especially for deep networks.

- $|1-f(x+v)-f(x)|<\epsilon$ is not a good criterion for poisoning, for example, $f(x+v)=f(x)=0.5$ corresponds to $\epsilon=0$, but this case is not considered as a successful poisoning.

- In Section 3.4, actually, I do not see any close connection between the concept of Trojan Twin and the proposed Algorithm 1. For me, Algorithm 1 is just constructing universal poisoning perturbation on an ensemble of adversarially trained models, which is nothing new beyond [1]+[2].

- Why the poisoning trigger in Algorithm 1 is preferred to be universal, rather than input-dependent? Besides, do the baselines in experiments exploit the same knowledge from adversarially trained models as done in the proposed method?


Refs:

[1] Universal Adversarial Perturbations. CVPR 2017

[2] Ensemble Adversarial Training: Attacks and Defenses. ICLR 2018

**Summary Of The Paper:**

This paper proposes the concept of Trojaned Twin Model, which provides an interesting view w.r.t. the clean model and its trojaned counterpart. In practice, a universal poisoning perturbation is craft on an ensemble of adversarially trained models, and imposed onto the training set for poisoning. Empirical evaluation are done on CIFAR-10, GTSRB and ImageNet.

**Summary Of The Review:**

I like the idea of Trojaned Twin Model, which is interesting and formally defined. However, the bound in Theorem 1 seems impractical, and the connection between the theoretical analyses and the proposed Algorithm 1 is loose.

---

> ### Author Response · Authors · 2022-11-19
> **Response to Reviewer YzEQ - Part II**
>
> __Q4-5:__ "Besides, do the baselines in experiments exploit the same knowledge from adversarially trained models as done in the proposed method?"
> >__Ans4-5:__ Thanks for pointing this out, you are correct and we should be more explicit on this point. There is no natural way for all baselines to exploit pre-trained networks given their original design. It requires non-trivial adaptation for these baselines to use adversarially pre-trained models like us.

---

> ### Author Response · Authors · 2022-11-19
> **Response to Reviewer YzEQ - Part I**
>
> __Q4-1:__ As to Theorem 1, there is $|1-f^*( x_{bad} +  v) - f^*(x_{bad})| \leq \varepsilon$. For a practical distribution $\mu(\boldsymbol x)$, if $\epsilon$ is small (satisfying the definition of Trojan Twin), then $x_{bad}$ is usually located at low-probability regions. So consequently, $\mu(\mathcal{B}(x_{bad},\varepsilon,\varepsilon/(4\beta)))$, i.e., the ratio of successful trojan  would be quite small. Besides, the practical model family usually has a large value of Lipschitz constant $\beta$, especially for deep networks.
>
> >__Ans4-1:__ We agree with the reviewer that if $x_{bad}$ is located in some low-mass region the bound could be too loose. However, we believe for a family of DNN models, there will be plenty of $x_{bad}$, which is observed in [1].
> Indeed, several observations have been made in [1] that there exists some common direction $\boldsymbol v$ which can simultaneously adversarially attack many $x_{\rm bad}$  that are distinct from each other.  Instead of considering a single bad data $x_{bad}$, one can take the union bound of a family of bad data introduced by the trigger, which could improve the sample complexity. We also agree with the reviewer on the comments about the large Lipschitz constant of the DNN family. We believe considering a family of hypotheses with relatively smaller Lipschitz Constant, e.g., family of DNN with Lipschitz regularization, could improve the bound in theorem 1. We believe this observation could also be of great practical interest.
>
> [1] Universal Adversarial Perturbations CVPR 2017
>
> __Q4-2:__ $|1-f(x+v)-f(x)|$ is not a good criterion for poisoning, for example, $f(x)=0.5$  corresponds to $\varepsilon=0$, but this case is not considered as a successful poisoning.
> >__Ans4-2:__ Good point. The criterion $|1-f(x+v)-f(x)|$ is motivated from changing the classification result from 'correct' class to 'wrong' class, while maintaining the uncertainty of classifier. There are pros and cons of this criterion. For example, maintaining the uncertainty avoids making TTM over-confident on hard/indistinguishable data (the $f(x)=0.5$ case), thus robust against uncertainty type inspection methods. As pointed out by the reviewer, this criterion might admit TTM that is too conservative, as the output of corrupted examples are not pushed toward extreme value $\{0,1\}$. We believe in applications other than classification where the continuous outputs are not discretized, this criterion might not be a good criterion.  We look forward to extending our results to more general criterion on other tasks in future work.
>
> __Q4-3:__ "Why is the poisoning trigger in Algorithm 1 preferred to be universal, rather than input-dependent?"
> >__Ans 4-3:__  This is determined by the nature of the Trojan attack problem. In Trojan attack, attackers require a unique trigger to work not only on Trojaned examples in the training database but to generalize to work on more unseen testing data, such that attackers can easily change the classification result of any inputs by simply overlaying the trigger on them.
>
> __Q4-4:__ "In Section 3.4, actually, I do not see any close connection between the concept of Trojan Twin and the proposed Algorithm 1. For me, Algorithm 1 is just constructing universal poisoning perturbation on an ensemble of adversarially trained models, which is nothing new beyond [1]+[2]."
> >__Ans4-4:__  Our theorem and algorithm are closely coupled to produce a trojan twin model with guaranteed nice properties. Our theorem is very powerful; it suggests the existence of a universal trigger such that any model trained with the trigger will automatically become a trojan twin. Our algorithm 1 finds such a trigger, and thus is theoretically guaranteed to produce a trojan twin.  In other words, the output of algorithm 1 automatically inherits all the nice properties of Trojan Twin: high clean accuracy and ASR, stealthiness, etc. These properties are not guaranteed for the output of any other attacking methods, to the best of our knowledge.
> To find the universal trigger, Algorithm 1 does use techniques resembling [1] and [2]. But this should not lower the quality of our work. The innovation of our method is mainly the perfect marriage between the theorem and the algorithm. We actually consider the relative simplicity of our algorithm a strength; it is easier to tune and generalizes well.

---

### Official Review · Reviewer_8fMD · 2022-10-24

**Confidence:** 4
**Correctness:** 2
**Technical Novelty And Significance:** 2
**Empirical Novelty And Significance:** 2
**Recommendation:** 3

**Clarity, Quality, Novelty And Reproducibility:**

The paper is in general easy to read for a person with a background on the ML backdoor problem, but there are some typos (e.g., "is very closed to"), and missing information (e.g., clean model set, data normalization to understand "L2 distance less than 10"). Due to the missing information, reproducibility is low despite the simple algorithm.

**Strength And Weaknesses:**

Strengths.
- S1: The paper tries to prove the existence of a backdoored model.
- S2: The proposed approach is compared to 5 baselines on three datasets.

Weaknesses.
- W1: The empirical performance doesn't show significant improvement. Actually, the performance is similar or worse than some of the existing approaches.
- W2: Neural networks are known to be a universal function approximator, and thus the existence of such a model doesn't seem to add much value, especially with adversarial examples with universal perturbation.
- W3: The approach is already known in both backdoor and evasion scenarios. Also, this type of optimization is commonly used in many attacks, and the approach is a direct adoption of universal perturbation, which has been also used for poisoning. Among many, a quick search shows "Detecting Backdoor in Deep Neural Networks via Intentional Adversarial Perturbations," Xue et al., 2021; "Hidden Trigger Backdoor Attacks," Saha et al., 2020; "Invisible and Efficient Backdoor Attacks for Compressed Deep Neural Networks," Phan et al., 2022; "AdvDoor: Adversarial Backdoor Attack of Deep Learning System," Zhang et al., 2021.
- W4: The experimental setup is unclear, and there are some modifications to common tasks which makes it harder to compare. For example, the proposed attack uses a set of clean models which is not specified, and can be used as an unfair advantage of the proposed attack, especially if the model has the same architecture or is trained on the same dataset as the victim model.

**Summary Of The Paper:**

This paper proposes a method to backdoor a neural network so that it classifies certain samples containing a trigger as the attacker intended. The proposed approach assumes to have a set of surrogate models to generate a universal perturbation as done in an evasion attack, and take the universal perturbation as a backdoor. The approach is empirically compared to baselines.

**Summary Of The Review:**

This paper revisits a known property of the backdoor problem. The empirical evaluation is unclear, and the improvement over the existing approaches is not significant. The model set is not explained which is very important in understanding the advantage and limitations of the proposed approach. The related work considers some of the existing optimization-based approaches model-based (the attacker has complete control of the training pipeline and deliver the poisoned model), which I believe is an oversimplification, as many optimization-based approaches use a surrogate model similar to the clean model set used in this paper to generate a poisoned training dataset. In summary, this paper does not seem to show a clear advantage of the proposed approach, has an unclear experimental setup, and misrepresent existing approaches.

---

> ### Author Response · Authors · 2022-11-19
> **Response to 8fMD - Part II**
>
> __Q3-4:__ "The experimental setup is unclear"... "uses a set of clean models which is not specified"..."if the model has the same architecture or is trained on the same dataset"..."missing information such as L2 distance less than 10"
>
> >__Ans3-4:__ We believe this must be a misunderstanding.  We have improved related presentations in the paper.
> We use 5 adversarially-pretrained models to search for the universal Trojan trigger. Each model in the pool is initialized differently from those used in the attacking experiment. In the paper, we use the same architecture for the pools and for the target model. The dataset is set to be the same under the Trojan attack setting (the attacker either delivers the data or controls the data).
>
> >We constrain the L2 norm of triggers to be smaller than 10 for all baselines. During the adaptive optimization, if the L2 norm of a trigger goes above 10 then we scale it to have norm 10. This is to guarantee the same trigger budget for all baselines.
>
> >To resolve the concern on architecture used by pretrained models, here we present the transferring attack results in Table 2-3 where we search the trigger with architecture A and attack model in architecture B. For example, if we inject triggers found with VGG16 to attack a ResNet18, we can get similar positive performance as in the original paper.
>
> >Table 2. Transferring Attack - Classification Accuracy on Clean Images
> | Dataset | Pooled Model | Target Model | Ours (in the paper) | Ours (use different architecture) |
> |---------|--------------|--------------|---------------------|-----------------------------------|
> | CIFAR10 | VGG16        | ResNet18     | 0.908$\pm$0.002     | 0.907$\pm$0.004                   |
> | CIFAR10 | ResNet18     | VGG16        | 0.904$\pm$0.001     | 0.905$\pm$0.001                   |
> | GTSRB   | VGG16        | ResNet18     | 0.912$\pm$0.002     | 0.914$\pm$0.050                  |
> | GTSRB   | ResNet18     | VGG16        | 0.946$\pm$0.009     | 0.939$\pm$0.004 |
>
> >Table 3. Transferring Attack - Attack Successful Rate
> | Dataset | Pooled Model | Target Model | Ours (in the paper) | Ours (use different architecture) |
> |---------|--------------|--------------|---------------------|-----------------------------------|
> | CIFAR10 | VGG16        | ResNet18     | 0.994$\pm$0.000     | 0.994$\pm$0.001                   |
> | CIFAR10 | ResNet18     | VGG16        | 0.994$\pm$0.000     | 0.994$\pm$0.000                   |
> | GTSRB   | VGG16        | ResNet18     | 0.978$\pm$0.000     | 0.976$\pm$0.001                   |
> | GTSRB   | ResNet18     | VGG16        | 0.976$\pm$0.004     | 0.977$\pm$0.001                   |
>
> __Q3-5:__ The related work considers some of the existing optimization-based approaches model-based (the attacker has complete control of the training pipeline and deliver the poisoned model), which I believe is an oversimplification, as many optimization-based approaches use a surrogate model similar to the clean model set used in this paper to generate a poisoned training dataset.
>
> >__Ans3-5:__ We follow this recent comprehensive survey [1] to categorize different schools of work. We have improved the literature discussion accordingly.

---

> ### Author Response · Authors · 2022-11-19
> **Response to 8fMD - Part I**
>
> __Q3-1:__ "The empirical performance doesn't show significant improvement. Actually, the performance is similar or worse than some of the existing approaches."
>
> >__Ans3-1:__ Thank you for raising this question. We have improved our paper and hope that we can avoid further misunderstanding. Indeed, given the fact that both the Accuracy and ASR have reached a saturated level, we believe the improvement of the proposed method is quite significant from a statistical perspective. Please refer to response Ans1-2 in the reply to Reviewer DxEo for more detailed discussion.
>
> __Q3-2:__ "Neural networks are known to be a universal function approximator, and thus the existence of such a model doesn't seem to add much value, especially with adversarial examples with universal perturbation."
>
> >__Ans3-2:__  We would like to re-highlight that we are not only proving the existence of a trojan twin. More importantly, we show that one is guaranteed to obtain the trojan twin very conveniently, by finding a universal trigger and poisoning the training data with it. Using VC (Vapnik–Chervonenkis) toolkit, we prove sample size conditions for the training with poison data. Notably, we also showed that the trojan behavior is generalizable to the inference stage, as we studied the trojan behavior at the distribution level.  To the best of our knowledge, this is the first theoretical study of Trojan models at the distribution level. All these properties are beyond what a universal function approximator can offer.
>
> __Q3-3:__ "The approach is already known in both backdoor and evasion scenarios..." "Also, this type of optimization is commonly used in many attacks"... "Among many, a quick search shows..."
>
> >__Ans 3-3:__ We believe it's a misunderstanding. Although all listed works promote the adaptive trigger instead of static ones, their fundamental mechanisms are significantly different from each other. Furthermore, all listed attack methods mainly focus on heuristic __white-box__ attacks in which one can control the training. Whereas, our method is a __black-box__ attack with __theoretical guarantees__; we only generate poisoned samples and do not interfere with training.
>
> >Specifically, these attack works assume having the access to apply regularization during training, such as hidden representation restriction in (Saha et al.,2020), activating level restriction in (Phan et al., 2022) and adversarial attack to the target model in (Quan et al.,2021).  While we don't assume any access to the Trojan model. We only search for a specific trigger and insert it into the data. We prove that training with this trigger can give us a Trojan's twin model that is close to the clean model. The architecture used to find the Trigger does not have to be the same as the target architecture.

---

### Official Review · Reviewer_HyKs · 2022-10-25

**Confidence:** 4
**Correctness:** 3
**Technical Novelty And Significance:** 4
**Empirical Novelty And Significance:** 4
**Recommendation:** 6

**Clarity, Quality, Novelty And Reproducibility:**

This paper is clear in writing. The results are novel as far as I know. Relation with previous work is also adequately explained.

**Strength And Weaknesses:**

**Strength**

The paper is technically sound. It provides both theoretically and empirical insights to the problem of backdoor attack.

**Weakness**

I have some questions regarding the technical details of the paper.

1) The theoretical bounds in Theorem 1 and Proposition 1 are somewhat disjoint from the experiment. The experiment checks the existence of universal backdoor trigger, but does not directly check the coefficient of these bounds. While finding the coefficient may be hard for real data, could you provide a toy example for, say, two Gaussian distributions and give the audience a sense of how big the coefficients are?

2) In Figure 2, the horizontal axis is the filter pruning ratio. Could you provide some references on how the pruned filters are selected? Does the curve shape in Fig 2 still hold if a different set of filters are pruned?

3) In Table 3, the mean AIV of your method is >2 against ABS. Why do you use a t-test to show the evasiveness instead of removing all training examples with AIV>2 and then retrain the model on the remaining samples?

**Summary Of The Paper:**

This paper investigate the existence of trojan model for backdoor attack --- a model that is in theory similar to the Bayes optimal yet performs poorly under some universal backdoor trigger. The paper formalizes its notion, proves its conditional existence and show an attack algorithm that generates universal backdoor triggers for real-world tasks.

**Summary Of The Review:**

Overall, this paper provides solid theoretical and empirical results for investigating the existence of trojan twin model and universal backdoor trigger. I'm giving a weak accept for now, and willing to raise once my questions are addressed.

---

> ### Author Response · Authors · 2022-11-19
> **Response to Reviewer HyKs**
>
> __Q2-1:__ "The theoretical bounds in Theorem 1 and Proposition 1 are somewhat disjoint from the experiment. "..."The experiment checks the existence of a universal backdoor trigger but does not directly check the coefficient of these bounds. "..."could you provide a toy example for, say, two Gaussian distributions..."
>
> >__Ans2-1:__ We agree that a tight connection between theoretical bounds and experiments could be helpful. As this work is the first to study statistical error bounds in Trojan Attack, it only covers sufficient conditions for low statistical error. The tightness of the bound, which requires information theoretic lower bound construction, remains open. We also believe that toolkits like the local Rademacher average could further improve the dependence on $\varepsilon$, toward a minimax optimal error rate.  We believe minimax optimal statistical error bound would be an important step toward obtaining a tight coefficient. We look forward to these exploration in the future.
>
> __Q2-2:__ "In Figure 2, the horizontal axis is the filter pruning ratio. Could you provide some references on how the pruned filters are selected? Does the curve shape in Fig 2 still hold if a different set of filters are pruned?"
>
> >__Ans2-2:__ We use the unstructured L1 pruning from [1], where for each layer we prune filters with the lowest L1 norm. We perform such pruning for all layers.
>
> >As the reviewer suggested, we also provide the global pruning result, where we prune the filter that has the smallest L1 norm among all convolutional layers instead of doing it in a stratified manner. With this pruning method, it is possible some layers can be totally removed with an  increased pruning ratio. We present the added experiments in appendix Figure-3.  The results follow a similar pattern as layer-wise pruning..
>
> [1].Hao Li, Asim Kadav, Igor Durdanovic, Hanan Samet, Hans Peter Graf, Pruning Filters for Efficient ConvNets, ICLR, 2016
>
> __Q2-3:__ 'In Table 3, the mean AIV of your method is >2 against ABS. Why do you use a t-test to show the evasiveness instead of removing all training examples with AIV>2 and then retrain the model on the remaining samples?'
>
> >__Ans2-3:__ Thanks for pointing this out. We admit that it is a confusing presentation for the AIV metric. We have improved the explanation of this metric in the paper. For each output neuron in each suspicious model, we could calculate an AIV value. Following the setting of [2], for a Trojaned model, we record the AIV given by the target class neuron. Then for each of these 10 trojan models, we can record the AIV of their target class. If a Trojaned model has an AIV value larger than 2, it is considered detected. We perform t-tests to statistically verify if each detection algorithm can consistently detect enough number of Trojan out of these 10 sample networks.

---

### Official Review · Reviewer_DxEo · 2022-10-26

**Confidence:** 4
**Correctness:** 3
**Technical Novelty And Significance:** 2
**Empirical Novelty And Significance:** 2
**Recommendation:** 5

**Clarity, Quality, Novelty And Reproducibility:**

This paper is a little bit hard to follow considering the usage of many informal expressions. The quality and novelty are relatively good. I did not check the reproducibility.

**Strength And Weaknesses:**

# Strength
1. Investigating backdoor attacks under adversarial training is relatively new.
2. The theoretical results seems interesting.


# Weakness
1. The claim that "existing backdoor attacks cannot resolve adversarial training" is questionable. The empirical evidences provided by the author do not fully support this claim.
2. The improvement of the proposed method is relatively marginal.

**Summary Of The Paper:**

This paper studies the backdoor adversarial attack. It theoretically discuss the existence of an attacking counterpart model and corresponding design an attacking algorithm. The proposed method is evaluated on benchmark datasets.

**Summary Of The Review:**

As in Strength vs Weakness

---

> ### Author Response · Authors · 2022-11-19
> **Response to Reviewer DxEo**
>
> __Q1-1:__ The claim that "existing backdoor attacks cannot resolve adversarial training" is questionable. The empirical evidence provided by the author does not fully support this claim.
> >__Ans1-1:__  We believe this is a misunderstanding. We have improved the presentation and hope that it can clarify our empirical results. Our experiment results actually show that adversarial training mitigates the effect of Trojan attack. If we look at Table 4-5 in the appendix, where we don't use adversarial training, both ASR and ACC of all attack methods are consistently much better than attacking against adversarial training (See Table 1-2).
>
> __Q1-2:__ “The improvement of the proposed method is relatively marginal.”
> >__Ans1-2:__ Thank you for raising this question. Indeed, considering the fact that both the Accuracy and ASR have reached a saturated level, the improvement of the proposed method is quite significant. To rigorously show the significance of improvement, we present following summary statistics:
>
> >In Table 1-2 in the main paper and Table 6-9 in the appendix, there are in total 28 sets of experiments. We performed two sample t-tests between the top 2 methods. This time, we only highlight a method if it is the unique best method without being tied with another one. Out of 28 experiments, the number of times each method to be the unique best method is presented in Table R1. We can see that our method stands out more times compared to all the baselines. We will add this clarification to the future revision.
>
> >Table R1. Number of Times to be the Unique Best Method
> | Method | BadNet | SIG | REF | WaNet | IMC | Ours |
> |--------|--------|-----|-----|-------|-----|------|
> | Times | 1      | 1   | 0   | 0     | 0   | 10   |

---

### Author Response · Authors · 2022-11-19
**Overall Response**

We sincerely thank all reviewers for their time and valuable suggestions. We have improved our manuscript based on constructive feedbacks from reviewers (changes highlighted in blue). We will address each reviewer's concerns separately.

---

### Decision · Program_Chairs · 2023-01-20

**Decision:**

Reject

**Justification For Why Not Higher Score:**

The improvements in the experiments are quite marginal.

**Justification For Why Not Lower Score:**

N/A

**Metareview: Summary, Strengths And Weaknesses:**

The paper proposes a novel concept of trojaned twin model to characterize the clean model and the model that is being back-doored. They then use a universal perturbation as a trigger for real-world tasks. Despite proposing an interesting concept, the empirical results of this paper are weak as pointed out by several reviewers. We therefore decide to reject the paper and encourage the authors to improve the empirical section in the future submission.